# Wireless Sensor Network Energy Model and Its Use in the Optimization of Routing Protocols

**Carolina Del-Valle-Soto [1,*]** , **Carlos Mex-Perera [2]** , **Juan Arturo Nolazco-Flores [3],**
**Ramiro Velázquez [4]** and **Alberto Rossa-Sierra [1]**

[1] Facultad de Ingeniería, Universidad Panamericana, Álvaro del Portillo 49, Zapopan, Jalisco 45010, Mexico; lurosa@up.edu.mx
[2] ITAM, Rio Hondo 1, Ciudad de México 01080, Mexico; carlos.mex@itam.mx
[3] Tecnologico de Monterrey, Campus Puebla, Vía Atlixcáyotl No. 5718, Reserva Territorial Atlixcáyotl, Puebla 72453, Mexico; jnolazco@tec.mx
[4] Facultad de Ingeniería, Universidad Panamericana, Josemaría Escrivá de Balaguer 101, Aguascalientes 20290, Mexico; rvelazquez@up.edu.mx
*   Correspondence: cvalle@up.edu.mx; Tel.: +52-33-13682200 (ext. 4245)

**Abstract:** In this study, a Wireless Sensor Network (WSN) energy model is proposed by defining the energy consumption at each node. Such a model calculates the energy at each node by estimating the energy of the main functions developed at sensing and transmitting data when running the routing protocol. These functions are related to wireless communications and measured and compared to the most relevant impact on an energy standpoint and performance metrics. The energy model is validated using a Texas Instruments CC2530 system-on-chip (SoC), as a proof-of-concept. The proposed energy model is then used to calculate the energy consumption of a Multi-Parent Hierarchical (MPH) routing protocol and five widely known network sensors routing protocols: Ad-hoc On-demand Distance Vector (AODV), Dynamic Source Routing (DSR), ZigBee Tree Routing (ZTR), Low Energy Adaptive Clustering Hierarchy (LEACH), and Power Efficient Gathering in Sensor Information Systems (PEGASIS). Experimental test-bed simulations were performed on a random layout topology with two collector nodes. Each node was running under different wireless technologies: Zigbee, Bluetooth Low Energy, and LoRa by WiFi. The objective of this work is to analyze the performance of the proposed energy model in routing protocols of diverse nature: reactive, proactive, hybrid and energy-aware. Experimental results show that the MPH routing protocol consumes 16%, 13%, and 5% less energy when compared to AODV, DSR, and ZTR, respectively; and it presents only 2% and 3% of greater energy consumption with respect to the energy-aware PEGASIS and LEACH protocols, respectively. The proposed model achieves a 97% accuracy compared to the actual performance of a network. Tests are performed to analyze the consumption of the main tasks of a node in a network.

**Keywords:** energy consumption; routing protocols; performance parameters; Wireless Sensor Network (WSN)

## 1. Introduction

Presently, houses, buildings, parks and, cities in general, involve several electronic devices working with different wireless technologies. The type of application determines the appropriate combination of standards and protocols to be used. One of the main characteristics of these wireless devices is their power requirements.

Wireless Sensor Networks (WSNs) involve devices characterized by small nodes, low energy consumption, limited battery life, low task processing, and low storage capacity. These self-configuring networks are easy to implement and to deploy. In these networks, communications through channels with multiple interferences and computing capabilities to operate at low energy are assessed. Sensor networks should exhibit an optimal performance with reduced delays and provide reliable information with a minimum energy consumption in order to provide valuable information for long periods [1]. However, energy consumption may become a major issue because of the low-battery power. The life span of the nodes should be as long as possible to avoid constant human intervention due to the harsh environment of some of their applications, such as in the study of natural behavior, risk areas, medical industry, domotics, agriculture, battlefields, and home networks [2].

Considerations on energy consumption are critical in sensor networks because their implementation must be simple, enduring, and resilient to topology or configuration changes. All these factors significantly influence the energy expenditure of a network and are represented by its performance parameters.

In this study, an efficient energy model for sensor networks is implemented with the aim of quantifying energy consumption during the execution of the main tasks of a node within a network [3]. The types of energy considered by this model correspond to the following phases of operation: turning on, channel auditing, receiving packets, sending packets, switching activities, microcontroller processing, and turning off. The novelty of this model is that it features a simple design scheme capable of characterizing energetic behavior against possible network anomalies in which consumption levels exceptionally increase, as it can be seen in [4]. In the proposed model, energy is considered to be an indicator of a typical behavior for a network exposed to interference attacks. In addition, it is an easy scheme to implement in a node, which provides reliable responses to modifications in node behavior. In this way, it is also possible to optimize energy consumption for specific node activities as well as for the overall network performance. The model allows for scalability and demonstrates the main reaction modes of a network node.

This paper intends to test the proposed energy model and observe its repercussions on proactive and reactive sensor network protocols. The analysis is described quantitatively by observing the performance metrics that would positively or negatively affect such model. This is where the contribution of this work becomes relevant: the proposed model quickly shows changes in the network performance, its implementation is simple, and it does not represent higher processing consumption. The proposed model is then compared against the performance of network sensors under some widely known protocols: Ad hoc On demand Distance Vector (AODV) [5], Dynamic Source Routing (DSR) [6], ZigBee Tree Routing (ZTR) [7], Low Energy Adaptive Clustering Hierarchy (LEACH) [8] and, Power Efficient Gathering in Sensor Information Systems (PEGASIS) [9]. These protocols will also be compared against the Multi-Parent Hierarchical (MPH) routing protocol proposed, designed, and implemented by the authors in a previous work [10].

There are also other protocols analyzed in the literature, such as Cluster-based Energy Efficient Location Routing Protocol (CELRP), which is a hierarchical protocol with nodes distributed in clusters and arranged in quadrants. Each quadrant contains two clustering, which would be like the master nodes, and other nodes transmit data with two hops data transmission. Another similar protocol is Position Responsive Routing Protocol (PRRP), which is more energy efficient. This protocol makes a choice of the cluster head based on distance from the sink, energy level, and the average distance of neighboring nodes from the candidate master node. PRRP is similar to the LEACH protocol in which any node can communicate with the sump and the data transmission mechanism is the time-based schedule. In PRRP, the number of nodes of the branches of the hierarchical tree and the distance from the non-leaf node is smaller compared to LEACH and CELRP. This makes energy conservation candidate for optimization [11]. The PRRP protocol dramatically increases data transfer and provides a better solution to the routing problem focused on energy efficiency, due to the efficient selection and distribution of gateways. Another important protocol to mention is the Energy-Efficient data

Routing Protocol (EERP) for WSN [12], which selects a set of good roads, and chooses the one based on the node state and the road cost function. In EERP, each node has several neighbors through which packets can be routed to the base station. A node bases its routing decision on two metrics: status and cost function.

In this study, AODV, DSR, ZTR, LEACH, and PEGASIS are quantitatively compared and assessed based on several efficiency metrics that analyze how these routing protocols optimize energy through various schemes in order to find the best routes in the shortest possible time. As the hierarchy algorithms, such as the ZTR, denote simple and fast routing that reduce network overloads, they are reliable and have a distributed addressing scheme that only permits neighbor tables, not long, and elaborated routing tables. The performance of WSN is closely related to that of the routing protocol, because routes can vary dynamically over time. Energy-aware protocols such as LEACH and PEGASIS seek to increase the lifetime of the network. They propose to find sub-optimal paths to allow a more equitable distribution of the network's energy consumption. Hierarchical protocols such as ZTR and MPH have advantages in terms of scalability and efficiency in communications. Particularly for WSN, nodes with higher energy can be used to process and send information, while those with lower energy are used to monitor the environment and send the information to the node with greater energy capacity. Finally, proactive type protocols, which establish routes before there is a real traffic demand, are suitable for real-time traffic, since they have low latency; however, they waste bandwidth due to periodic updates and they are not energy efficient. The MPH protocol is a hybrid protocol, i.e., it is a combination of reactive and proactive nature protocols. The AODV and DSR protocols are two protocols widely recognized in WSN for their rationality and the ZTR protocol is a proactive protocol par excellence. Therefore, we believe that the comparison between protocols of natures of the same type is relevant. In addition, we wanted to complement our study with energy-aware protocols because the main objective of this work is to demonstrate and analyze the higher energy costs in the sensors, according to the type of tasks they perform on the network.

This paper proposes a simple energy model, which quickly shows changes in the network performance, its implementation is simple, and it does not represent higher processing consumption. In the WSN literature, there are few energy models [13,14] and some energy-aware protocols that seek to optimize the energy of networks. The need for an energy model that impacts the performance metrics of a network is an advantage that not all models exhibit. This indicates that we can know, according to each type of task that the node performs, what is the major and minor impact on parameters such as: resilience, overhead, packet retransmissions, listening retries to the communication channel, delay, and many others.

The energy consumption problem is not the same for all network nodes [15]. This is due to the fact that there are several collector nodes that cluster information around them. This coordinating collector node is robust, with its own energy supply and with greater processing resources than other nodes in the network. Consequently, this node has the capacity to process all the information gathered from the nodes of the network and subsequently, obtains results when assessing the information received. When there are one or more collector nodes and there are nodes nearby that forward all network traffic, they are more likely to exhaust faster their energy. This problem is known as the energy hole problem [16] and generates a high amount of packet losses, which will be represented by collisions. Unequal energy depletion causes the expiration time to unexpectedly generate loss of information from the network.

A possible solution to this problem, as per the literature, is the creation of groups to promote network scalability and the problem of zoning [17]. The analysis of the network areas is distributed in concentric rings to stimulate traffic between nodes as they approach their destination. The authors in [18] study variables such as constant bit rate, where the nodes are uniformly and randomly distributed. Performance parameters are analyzed to establish relationships between the different rings located near and far away from the collector node. In this study, the concept of node zones is used to observe energy repercussions based on accurate and reliable performance metrics that

directly influence energy expenditure and are presented as reliable evidence of network changes or anomalies. The link load imbalance problem is addressed through tree-type topology protocols aimed at minimizing packet delays and the number of jumps in the path to the root node or collector node. This imbalance is reflected as a non-uniform energy expenditure on the nodes. This failure is compensated in reactive protocols by adding more links, but this is prone to greater delays or outdated routes and their maintenance may become a challenge [19]. The work in [20] shows a method for balancing network traffic and ensuring uniform use of destination node routes. Nezhad et al. [21] propose a protocol in which the collector node has a global view of the network topology to maximize the life span of each node and the use of a load balancing algorithm to select the best routes.

Herein, some energy consumption performance metrics are proposed and used to compare the MPH routing protocol against other known sensor network protocols: AODV, DSR, ZTR, LEACH, and PEGASIS, all as per the IEEE standard 802.15.4/ZigBee. Node shutdown tests are performed to study how the network behaves according to the routing protocol implemented and its response capabilities [22]. The results of the simulation show how performance, reliability, and energy consumption are affected within the communications network. In this light, MPH is shown to be an efficient protocol as it presents the best performance among the protocols under evaluation. Compared to AODV, MPH routing exhibits a 26.9% decrease in overall energy consumption and a 41.2% increase in the protocol's ability to recover the topology after a failure event. In addition, an energy model for the CC2530 chip is proposed and used in the simulations of the four protocols aforementioned, resulting in a 16%, 13%, and 5% reduction in energy consumption for the MPH routing when compared to AODV, DSR, and ZTR, respectively. The proposed model allows us to determine that between MPH, LEACH and PEGASIS there is only a difference of 3% and 2% energy savings for the last two protocols. Thus, the model analyzes the energy impact of each type of energy for optimization of the algorithm in various protocols of the literature. These protocols and the energy model are implemented in an event simulator programmed in C++. The proposed energy model is implemented in the simulator for each routing protocol to observe the impact of the performance parameters and how they influence energy consumption in each protocol. This proposal resembles the MPH routing protocol [10], which is a hybrid that combines proactive and reactive protocols and energy conservation properties at the same time. It is used in this work as a protocol with hierarchical topology for optimizing the sending of information, maintaining a smaller number of routes, and monitoring them to determine whether they remain valid or have become obsolete [23].

Figure 1 shows a conceptual scheme for a hierarchical topology with a sink node or information collector. This scheme is widely used in WSN for low power consumption [24]. The problem described here is that energy is not distributed equally throughout the topology and there are different energy levels called crowns [25]. Energy increases in the nodes that form information bottlenecks. Therefore, in this work we classify the types of energy to establish compensations in the busiest nodes of the network. This scheme overviews the energy consumption of a many-to-one network. The coordinator or sink node is the sole destination of the network nodes (there may be more than one coordinator node), which makes it a single failure point. The nodes send traffic to the coordinator node. As the nodes are closer to the coordinator, they have to forward traffic from other network nodes. Therefore, with this scheme, we want to represent an ideal scenario of energy behavior of the nodes from the coordinator node to the nodes farthest from it. Being the darkest color the higher energy consumption nodes and the lightest, the lower energy consumption ones.

## 1.1. Motivation

This work presents a simple energy model based on the detailed analysis of the main types of energy consumed during the different tasks performed by sensor within a network while in active mode. This model is capable of performing a detailed breakdown of such types of energy according to the node hierarchy in the network considering at the same time both high and low consumption sources. This energy model is characterized by being simple and providing evidence of the energy

used by the main tasks of a node in the network. In addition, it reflects the impact of each type of energy on the performance metrics of a network, with which we could optimize routing protocols according to the conditions of the network and its local expenditure (at node level).

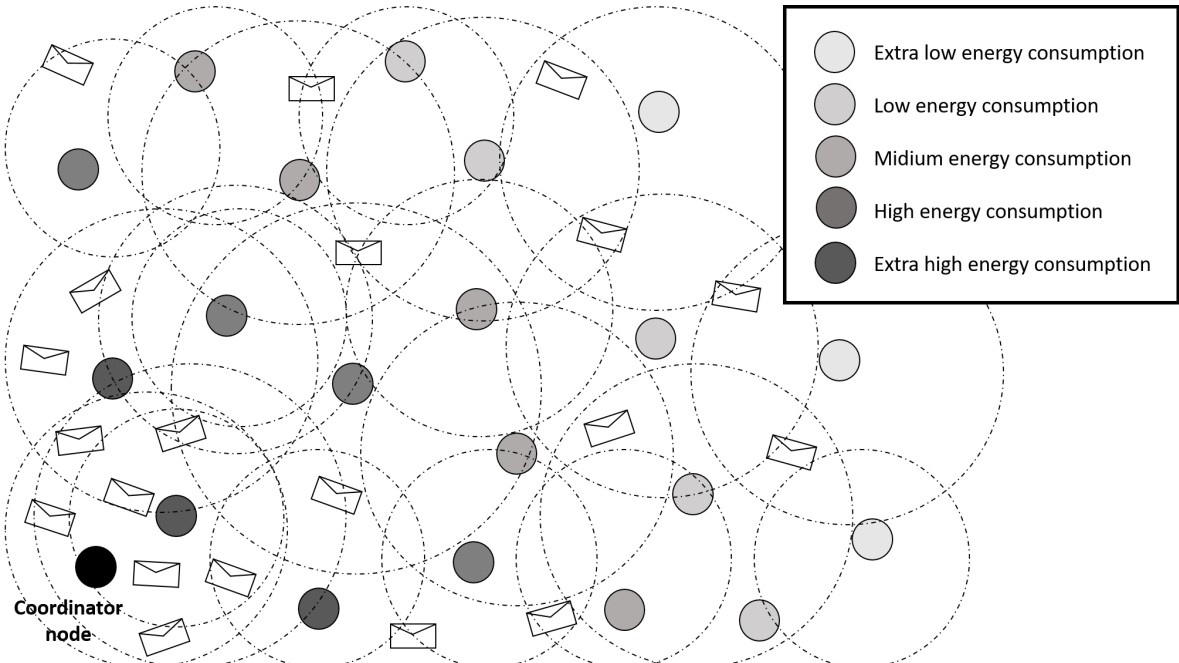

**Figure 1.** Conceptual energy consumption model in a hierarchical topology.

The results in this proposal are established by assigning a node hierarchy based on the traffic load due to the node's proximity to the coordinator node. The novelty in this proposal is the possibility of obtaining the skeleton of each energy expenditure in the network and establishing optimizations for higher consumptions. Furthermore, these network consumptions are classified by the proposed energy model according to their dependency in data packet sending. For example, if the unit's microcontroller is detected to be of high consumption, it can be replaced by a smaller one or for a chip with a better wireless technology. Similarly, if the consumption during data reception or transmission is high, it can be an indicator that the channel exhibits high interference and intermittent links. This way, this simple and easy to implement model intends to be the starting point for specific task node optimization methods and sleeping algorithms.

### 1.2. Scope of Routing Protocols in Sensor Networks

Most of the nodes in a sensor network have a limited power supply and do not have the ability to generate their own energy. Therefore, the design of energy efficient protocols is critical for the longevity of the network. A protocol for sensor networks must be configured in such a way that its operation does not require human attention. The IEEE 802.15.4 standard is used by virtually all wireless sensor devices today. Since a direct link between any node in the network and the coordinating node cannot necessarily be established, a multi-hop network topology and an algorithm are required to determine the route the messages follow. These are dynamic topologies with nodes that can stop operating due to physical failures or lack of batteries, with bandwidth restrictions, links with variable capacities, and equipment that can operate with energy restrictions. All these factors result in reconfigurations or unpredictable changes in the topology that handles the routing protocol. On the basis that many routes can communicate to a node with the base station or coordinating node, the objective of an energy-aware algorithm is the selection of those routes that maximize the lifetime of the network. In consequence, those routes composed of nodes that have greater autonomy are marked as preferred. Presently, some of the hierarchical protocols widely used in sensor networks are: LEACH (Low Energy

Adaptive Clustering Hierarchy) [26,27], PEGASIS (Power-Efficient Gathering in Sensor Information Systems) [28], TEEN (Threshold-sensitive Energy Efficient Protocols) [29], and PAMAS (Power-Aware Multi-access protocol with signaling) [30], among others.

LEACH is a cluster-based protocol that randomly selects a few nodes and treats them as master nodes. Its objective is to distribute the energy load evenly throughout the network. PEGASIS is an improvement of the LEACH protocol. Its main idea is to extend the life of the network by restricting node communication only to the closest neighbors and taking turns for communicating with the coordinating node. PEGASIS assumes that each node must be able to communicate with the base station directly. The TEEN protocol is used in applications where a rapid response is required against sudden changes in the parameters to be measured in the network. In TEEN, the nodes monitor the medium permanently, but the data transmission is sporadic. Since the transmission of messages consumes more energy than their capture, the consumption is lower than in proactive networks. PAMAS is a containment-based protocol where one of the main objectives is related to efficiency in energy consumption. This protocol adds separate channels for the control of RTS/CTS packets and data packets. "A busy tone" is attached to the signaling channel so that the nodes that do not participate in the communication turn their radio transceivers off. This way, PAMAS tries to avoid over-listening between neighboring nodes and does so by adding a second low power radio in their devices.

In communication networks, depending on the way the source creates a route to the destination, the routing protocols are classified into three groups: proactive routing protocols, reactive routing protocols, and hybrid protocols.

When nodes are under a reactive protocol, they ask for a single route as required. This implies a high latency for the first packet and some independence between the routes. Two common examples are the AODV and DSR protocols. These routing protocols are characterized by their reactive nature. That is, they both begin their search activity on demand routes. The difference between them is that DSR uses source routing, while AODV uses hop-by-hop routing by maintaining tables and sequence numbers in the destination nodes.

The AODV routing protocol is based on the routing efficiency of ad hoc wireless networks with a large number of nodes and uses a mechanism for route discovery in broadcast mode. AODV can transmit in unicast or multicast mode, uses bandwidth efficiently, and quickly responds to network changes avoiding network loops [5]. Each node in the network has a sequence number and a unique identifier in the network. This guarantees the absence of loops and avoids counting packets to infinity. To maintain and track routes to the neighbors, nodes periodically send *HELLO* messages. The nodes have a lifetime each time a node receives a packet from a neighbor. At this time, the neighbor's entry is updated in the routing table. If no entry has been defined for this neighbor, the node creates a new entry in the routing table. Therefore, information from the *HELLO* packet is used by neighbors to notify other nodes that the node itself is still active. This information is used by the neighbors to update the timers associated with that node or, alternatively, to disable the entries that are not responding. In fact, AODV maintains time-based states in each node's routing tables. A routing table entry expires if it has not been used recently. The timer function prevents using links which status has been marked as unknown for a long time. Some advantages of AODV are high reliability and low bandwidth costs. However, there are some disadvantages such as high complexity, additional calculations required, extra memory costs, and the fact that this protocol was designed to work in a network where there are no malicious nodes. In sum, it is not a secure protocol.

The DSR protocol is also a reactive protocol. This protocol routes from the source node including a header in the packets. This header indicates which nodes may need to be crossed to arrive at a destination since the originating node is responsible for calculating the complete route to the destination node. This process is called **Source Routing**. DSR does not require any periodic message. In this way, it reduces message overload. For this, when the origin node moves or the topology of the network changes, the algorithm perceives the modifications and adapts accordingly. In addition, DSR handles unidirectional links and asymmetric routes. Each node in the network has a cache memory that stores

all the routes obtained through the discovery processes from the node itself, which may consume slightly additional processing. If there is no current route to a specific destination, the node starts a reactive route discovery just like AODV.

The route table or route cache is constantly monitored to detect invalid routes and repairs them as the network topology changes. This process is called **Route Maintenance**. DSR features some advantages such as that nodes can obtain multiple routes to a specific destination but requesting only one route. DSR allows the network to be completely self-configurable, without a specific architecture or topology. In addition, it is a good choice in scenarios where the number of mobility nodes is reduced. This protocol adapts quickly to routing changes when a node is frequently moving and decreases network overload.

Next, we study a hybrid (proactive and reactive) routing protocol previously proposed, called MPH, which creates a hierarchical network logical topology where node hierarchy is proactively given by its location in the tree. Basically, the root hierarchy (sink or collector node) has the highest hierarchy. When a node has a hierarchy level, it can only have links from parents or children, directly connected. The hierarchical topology minimizes the number of hops and optimizes the routes to the coordinating node. The advantage of this protocol is that it combines features of both proactive and reactive nature and presents redundancy without losing simplicity in the algorithm. The coordinator can also send packets to the network nodes. An origin routing approach is adopted so that traffic is sent from the collector to any node, because the coordinator node has more resources and capacity than the rest. This means that information may be easily collected from the network, such as sensor-generated data, link quality metrics, neighbor node tables, and other variables that may be used to locate routes and for performance analysis and network optimization. If a node sends a packet to the destination or collector node, it searches for its parents in its Neighbor Table, selects a route, and sends the message. This process is repeated hop by hop until the destination. The sink node knows the complete topology of the network through frequent query packets at times previously established in the network nodes. These packets are sent through the hierarchical branches and request each node to send its information to the coordinator node. This protocol takes advantage of the proactive controlled route maintenance but combines the agility of having more than one route per node. This makes it more versatile and adaptable to other topologies.

ZTR is a simple protocol that establishes parent-child links with the nodes always carrying information to their parent. This protocol features a tree topology, is easy to implement, fast, and is proactive. ZigBee networks require at least one full-function device. That is, a robust device acting as a network coordinator, but the final nodes of the star may have low specifications to reduce costs. Before transmitting to a channel, the node must measure the energy level in a specific channel. This measurement only indicates if the channel is busy, but the node is not able to identify whether that energy corresponds to another device under the IEEE 802.15.4 standard. Here, the child node would be the one that most recently entered the network. The parent node is the node that has given the child access to the network. In this way, parent-child links are created, but each child can only have one parent. Some of the advantages of ZTR are balance between cost per unit, battery expenditure, and implementation complexity to achieve an appropriate cost-performance ratio for the application.

*1.3. Significance for Study of Energy in Wireless Sensor Networks*

To evaluate the network performance, one may consider parameters that evidence proper network operation directly influencing the energy consumption of each node. There are local and global parameters. Global parameters display the total energy costs for the network considering each type of energy for each specific activity. In contrast, local parameters provide total energy consumption rates for a single node. This energy depends on the location of the node within the topology regardless of how near or far they are located from the coordinator node and how much traffic is transmitted through it [31].

An energy-efficient routing protocol decreases the consumption of the nodes by routing data through paths that display the least amount of energy. There are some special mechanisms to achieve this goal such as optimization of jumps to the destination node, maintenance of optimal and valid routes, reduction of transmission delays, and reduction of packet retransmissions and attempts to listen to the channel [32,33]. In this paper, we assess the aforementioned aspects to compare against four sensor network protocols.

Concerning the communication channel, it is a factor that significantly influences the energy consumption because the protocol executes a series of listening attempts to determine whether the channel is already busy with other information packets. The carrier senses multiple accesses with the collision avoidance (CSMA/CA) protocol [34] and works as follows: first, a node begins listening to the wireless channel and if it is free, the node begins transmitting. If the wireless channel is not free, the node recalculates a random delay, waits, and listens again. We are using the MAC-level protocol used for all extensions of 802.15.4 (including the original version), which is the CSMA/CA that guarantees a high data rate. A network recognition is being carried out at all times to check the status of the channel (carrier detection). Only when free, data can be sent. In the 802.11 standard, the physical layer polls the energy level over the radio frequency to determine whether or not there is transmission. If the channel is busy, a random timer starts (with a maximum of five back off periods), the timer only discovers time with free channel, transmits when it expires, and finally, if it does not receive ACK, it increases the back off. This metric is known as CSMA/CA retries. If these CSMA/CA retries are frequent, the channel is busy most of the time. Consequently, there might be several collisions due to overload. In addition, when the wireless channel is permanently busy with information packets, there are many collisions and retransmissions of packets. This fact influences energy consumption because the nodes spend more time and capacity retransmitting over and over.

In a network layer, overloads are an important factor that influence energy consumption. The efficiency of the routing protocol may also be measured by the number of packets the protocol needs to route to its destination. A protocol with many control packets will contribute to packet collisions and overall performance reduction. In terms of route discovery, in all the protocols considered, the nodes exhibit capacity to know their neighbors. In AODV and DSR, the nodes update the routes as required. However, in ZTR and MPH routing, nodes periodically update routes. Known neighbors helps nodes to establish valid destination routes to forward messages and to reduce the number of retransmission packets required. The metric named 'valid routes' consists of paths that are not damaged and these routes are immediately available and ready to be used for sending information. If nodes are able to handle several valid routes, it is more likely that packets will be forwarded and will not be retransmitted continuously, thus saving energy resources.

Another important metric is found in the tables where the routes to different network destinations are located. These routes may either be valid or obsolete. Hence, their maintenance is essential to prevent loss of information within the network. Network energy consumption is directly related to the complexity in the administration of routing or neighbor tables. As sensors execute huge routing processes, energy consumption increases if these routes have not been properly updated. This is why it is also important to assess route delays; they are directly related to the number of jumps that a node takes to reach a destination.

This work seeks to further advance the study of performance metrics in the analysis of sensor networks. One of the most important contributions of this work is the relationship between metrics and their influence on energy consumption in a WSN, as well as the comparison of how these metrics perform in widely known routing protocols with respect to a protocol proposed by the authors. An important assessment of these parameters is that routing tables provide critical information about network reliability and the number of valid routes available to the nodes for sending information. These metrics will directly influence packet loss rates in the network as well as information delivery reliability. The aforementioned study is implemented in a sensor network for each routing protocol in which the proposed energy model is also implemented, as described in Table 1. This model is based

on the operation of the Texas Instruments CC2530 sensor [35] in active mode and stable conditions. Besides, for nodes under LoRa technology and WiFi, we use the model described in [36].

**Table 1.** Energy model base for each of the main tasks in a node [37].

|  | Voltage (mV) | Current (mA) | Time (ms) |
|---|---|---|---|
| Turn on mode | 120 | 12 | 0.2 |
| Microcontroller 32-MHz clock | 75 | 7.5 | 1.7 |
| CSMA/CA algorithm | 270 | 27 | 1.068 |
| Switching from reception to transmission | 140 | 14 | 0.2 |
| Switching from transmission to reception | 250 | 25 | 0.2 |
| Radius in reception mode | 250 | 25 | 4.1915 |
| Radius in transmission mode | 320 | 32 | 0.58 |
| Turn off mode | 75 | 7.5 | 2.5 |

Global parameters denote the energy rates that each node spends for all activities performed within the network. These types of energy include the energy consumed by the microcontroller, which is regulated by the time the sensor remains on; the CSMA energy used when the node is listening to the channel to determine whether packets may be transmitted or received and the node is executing the CSMA/CA algorithm; the switching energy consumed when switching between activities, i.e., changing from transmission to reception or vice versa; transmission and reception energy used to transmit or receive a packet, respectively; awakening energy used when the node turns on; and the shutdown energy consumed when the node turns off.

All of these energies will promptly report how nodes save or consume energy, as the case may be, together with the routing protocol. Each type of energy is provided as an overall network metric. In other words, each energy is incorporated to observe possible points of interest or zoning in the network.

Local energy balance depends on the proximity of a node to the destination node. In the scope of this work, the final destination is the coordinator or collector node. As a local aspect, the energy consumption at each node is calculated, which denotes consumption depending on the distance between the node and the coordinator. Thus, the nodes closest to the collector will send their packets and also retransmit packets coming from other nodes. For this reason, the relational position of a node within the network significantly matters.

## 2. Description of the Energy Model

We propose a simple energy model under the basic tasks or activities that nodes perform in the network [38]. Figure 2 conceptually describes this scheme. This model considers all the energy components that contribute to the overall energy consumption under active mode. First of all, a node remains at start time ($t_{ON}$) to turn on. Then, it takes a switching time ($t_{Switching}$) to change status before sending a packet to the medium. Here, the node first runs the CSMA algorithm using a CSMA time ($t_{CSMA}$). Next, the node transmits an information packet expending a transmission time ($t_{TX}$). Now, the node takes a switching time ($t_{Switching}$) to change activities, it remains inactive ($t_{Inactive}$) and changes task again. In addition, it takes a switching time ($t_{Switching}$) to start receiving information and reporting a reception time ($t_{RX}$). The node performs these activities as many times as it sends and receives information (messages) during the sampling period. Finally, the node turns off expending a shutdown time ($t_{OFF}$). All of this while the microcontroller remains in active mode. This process measures the energy required for each main node activity in the network. Depending on the task and the time a node takes to execute it, this corresponds to a given voltage and current, so that the total energy used by each node can be obtained for each of the activities run for the network based on the previous model [37].

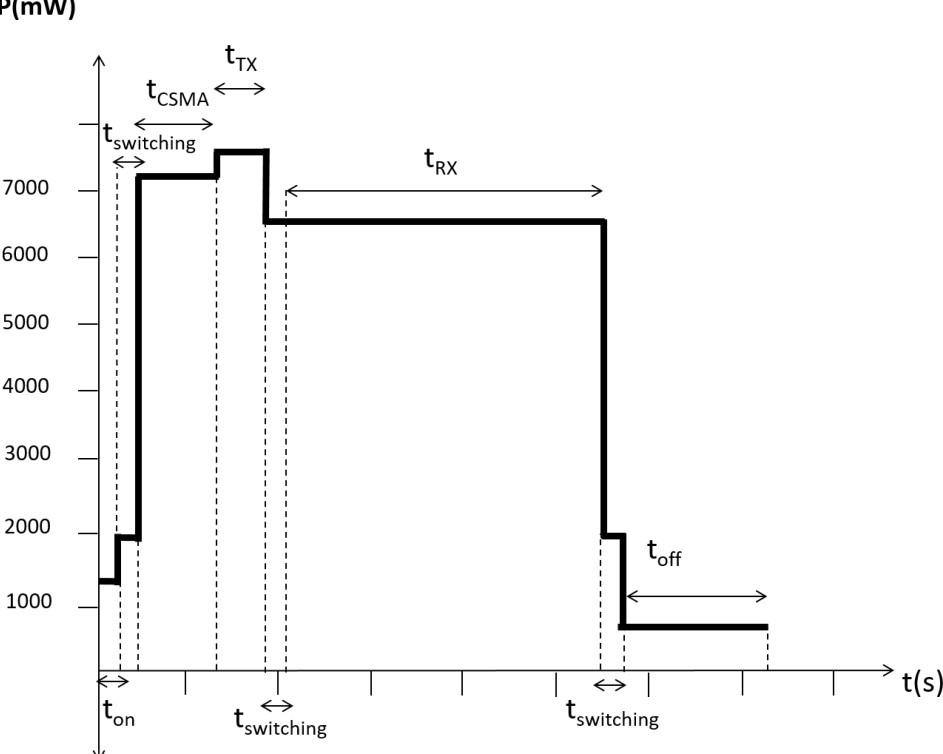

**Figure 2.** Conceptual energy scheme.

To compare the performance of the energy model, we used two types of sensors: CC2530 and CC2650 from Texas Instruments. We have coded the proposed model in a wireless sensor simulator programmed in C ++ based on the previously tested events [10]. The wireless module with antenna CC2530 nodes have 256 KB Flash and 8 KB of RAM sufficient for the implementation of widely known protocols in WSN. They have an IEEE 802.15.4 RF transmitter/receiver in 2.4 GHz high sensitivity (102 dBm). These sensors have four modes of operation in terms of energy savings and 21 general purpose pins, which cover the needs for connection of extra sensors. The CC2650 nodes are ultra-low power 2.4 GHz devices. An active and very low MCU and RF current and a low power mode current consumption offer excellent battery life. These devices contain a 32-bit ARM Cortex-M3 processor that operates at 48 MHz as the main processor and a set of extensive peripheral functions including an ultra-low power single sensor controller ideal for interconnecting external sensors.

Table 2 presents some results to verify the accuracy of the proposed energy model in predicting the total energy used by each node. We have taken as a basis the consumption for each type of energy proposed in the model. The described scenario is a configuration of 25 randomly and evenly distributed nodes and one of them is the coordinating or concentrating node. We left the nodes running from Monday to Sunday (seven days) on a university campus in an area of $500 \times 500$ m$^2$ under the conditions described in Table 3. We observed that the model described in Table 1 can be a reliable parameter to measure the general consumption of a wireless network of small to medium sensor size if the nodes are in active mode and the types of energies described give a faithful ideal of the main functions of a node in a network, taking approximately 4 to 5% margin of measurement error.

In this paper, an analytical model is proposed and exemplified by the operation parameters of the Texas Instruments CC2530 chip [35], which has a radio interface as per the IEEE 802.15.4 standard. This model shows the energy expended for each of the activities performed from the moment a node is added to a network and becomes part of it, listens to the channel, receives and sends messages, executes the link layer algorithms data, and changes states, ending with the energy consumed when it turns off and disconnects from the network.

**Table 2.** Energy model tested on two scenarios.

| Scenario | $E_{MC}$ (J) | $E_{ON}$ (J) | $E_{OFF}$ (J) | $E_{CSMA}$ (J) | $E_{Switching}$ (J) | $E_{TX}$ (J) | $E_{RX}$ (J) |
|---|---|---|---|---|---|---|---|
| Simulator for CC2530 | 9.6398 | 2.9342 | 14.1883 | 78.4582 | 7.4382 | 59.9173 | 264.1783 |
| Real network for CC2530 | 9.2578 | 2.5573 | 14.0123 | 78.9362 | 7.8625 | 59.1173 | 264.8746 |
| Simulator for CC2650 | 8.4677 | 2.3891 | 12.1903 | 77.3274 | 6.1248 | 56.4702 | 260.9471 |
| Real network for CC2650 | 8.4677 | 2.6891 | 12.1903 | 77.3274 | 6.1248 | 56.4702 | 260.9471 |

The energy used by the microcontroller depends on the node's operation mode. For example, techniques for turning nodes off reduce energy consumption by setting the microcontroller in idle mode for certain time intervals [39]. However, for this analysis, it is assumed that the mentioned SoC at each node operates in continuous active mode at 32 MHz (the microcontroller's clock frequency) to better study how energy consumption behaves under a specific routing protocol without the influence of techniques used for turning nodes off. Thus, the total energy used by the microcontroller will be given by (1):

$$E_{MC} = T_{MC} \times I_{MC} \times V_{MC}, \tag{1}$$

where $T_{MC}$ is the time (seconds) taken by the microcontroller unit to consume $V_{MC}$ (Volts) and $I_{MC}$ (Amperes).

The starting energy is estimated based on the voltage, current, and time nodes required to turn on and be ready for the network, as described in (2):

$$E_{ON} = T_{ON} \times I_{ON} \times V_{ON}, \tag{2}$$

where $T_{ON}$ is the time (seconds) that it takes to turn a node on and is given by $V_{ON}$ (Volts) and $I_{ON}$ (Amperes).

The model also describes the energy consumed when nodes turn off when the network time has ended (the sampling period). This energy is given by (3) and it is called shutdown energy.

$$E_{OFF} = T_{OFF} \times I_{OFF} \times V_{OFF}, \tag{3}$$

where $T_{OFF}$ is the time (seconds) that it takes to turn a node off and is given by $V_{OFF}$ (Volts) and $I_{OFF}$ (Amperes).

Switching energy is expended when the node changes from receiving to transmission mode or vice versa. It is given by (4):

$$E_{Switching} = T_{Switching} \times I_{Switching} \times V_{Switching}, \tag{4}$$

where $T_{Switching}$ is the time (seconds) that it takes a node to change from the reception mode to the transmission mode or vice versa and it is given by $V_{Switching}$ (Volts) and $I_{Switching}$ (Amperes).

The CSMA/CA algorithm states that each time a node intends to transmit, it first checks if the channel is free from other transmissions. If it is, then the node proceeds to transmit. Otherwise, it calculates a random waiting time before attempting to listen to the channel again. The energy consumed by the CSMA/CA algorithm is given by (5):

$$E_{CSMA} = T_{CSMA} \times I_{CSMA} \times V_{CSMA}, \tag{5}$$

where $T_{CSMA}$ is the time (seconds) in which a node computes the CSMA/CA algorithm and is given by $V_{CSMA}$ (Volts) and $I_{CSMA}$ (Amperes).

Transmission energy depends directly on the distance and interference. In this case, the nodes transmit their packets or forward packets from other nodes. Transmission energy is described as (6):

$$E_{TX} = P_{Length} \times T_{TX} \times I_{TX} \times V_{TX}, \tag{6}$$

where $P_{Length}$ is the length of the packet (bytes), $T_{TX}$ is the time (seconds) that it takes a node to send a byte and is given by $V_{TX}$ (Volts) and $I_{TX}$ (Amperes).

As with the transmission mode, a node expends receiving energy when it receives packets. This energy is determined using (7):

$$E_{RX} = P_{Length} \times T_{RX} \times I_{RX} \times V_{RX}. \tag{7}$$

where $P_{Length}$ is the length of the packet (bytes), $T_{RX}$ is the time (seconds) that it takes a node to receive a byte and is given by $V_{RX}$ (Volts) and $I_{RX}$ (Amperes).

The values of (1)–(7) are presented in Table 1.

Therefore, the total energy can be calculated with (8):

$$E_{Total} = E_{MC} + E_{ON} + E_{OFF} + E_{CSMA} + E_{Switching} + E_{TX} + E_{RX}. \tag{8}$$

The simple energy model shown by the aforementioned equations estimates the energy expended by the main tasks of a node within a WSN for any system, where those functions related to wireless communications are the most important from an energy standpoint.

This model can assess global and local energies. A generic case is presented to interpret the analysis of the model. The energy of the node $i$ is represented as $EnergyNode_i$, which is obtained by adding up the energy consumed by a node $i$ to perform each task along the active network. When a node connects to the network, it has consumed zero energy initially, so that $EnergyNode_i = 0$.

For this energy model, we segregate energies that depend directly on the number of packets transmitted or not. The energies related to the packets are: $E_{TX}$, $E_{RX}$, $E_{Switching}$ and $E_{CSMA}$. The energies that depend exclusively on node operation are: $E_{ON}$, $E_{OFF}$, and $E_{MC}$. Therefore,

$$EnergyNode_i = \overbrace{E_{TX_i} + E_{RX_i} + E_{Switching_i} + E_{CSMA_i}}^{\text{Dependent on packets}} +$$
$$\underbrace{E_{MC_i} + E_{ON_i} + E_{OFF_i}}_{\text{Independent on packets}}. \tag{9}$$

Now, to calculate the energy consumed by each node starting with packet-dependent energies, $E_{TX_i}$ will be the energy transmitted in the node $i$, so,

$$E_{TX_i} = (P_{Length} \times T_{TX} \times I_{TX} \times V_{TX}) \times (P_{TX_i} + P_{RTX_i}), \tag{10}$$

where $P_{TX_i}$ is the total number of messages transmitted by node $i$. $P_{RTX_i}$ is the total number of packets retransmitted by node $i$, because a packet receipts acknowledgment, ACK, was not received.

In addition, $E_{RX_i}$ will be the energy transmitted in the node i, so,

$$E_{RX_i} = (P_{Length} \times T_{RX} \times I_{RX} \times V_{RX}) \times P_{RX_i}, \tag{11}$$

where $P_{RX_i}$ is the total number of packets received by node $i$.

In the same way, $E_{Switching_i}$ is the switching energy consumed when a node changes from transmission to reception or vice versa. The $E_{Switching_i}$ energy for node i is given by (12):

$$E_{Switching_i} = (T_{Switching} \times I_{Switching} \times V_{Switching}) \times (P_{TX_i} + P_{RTX_i} + P_{RX_i}), \tag{12}$$

The term $(P_{TX_i} + P_{RTX_i} + P_{RX_i})$ represents the number of times there is switching or changing in state activity from the time the packet is sent to the time the packet is received or vice versa. Therefore, the best option is when a node sends a packet to a channel that is free and an acknowledgment ACK is

received. Then, $P_{TX_i} = 1$ and $P_{RX_i} = 1$. If the node receives the ACK, it does not retransmit the packet and $P_{RTX_i} = 0$. This switching operation is performed in two different stages: the node uses switching energy to change state, transmits the information, uses again switching energy, and receives the packet information.

Transmission and receiving energies are considered by (10) and (11), respectively.

For each packet transmitted, the node *i* executes the CSMA/CA algorithm. The corresponding energy consumed, $E_{CSMA_i}$ is given by (13):

$$
\begin{aligned}
E_{CSMA_i} = \quad & (T_{CSMA} \times I_{CSMA} \times V_{CSMA}) \times \\
& (P_{TX_i} + P_{RTX_i} + N_{RT_i}),
\end{aligned}
\tag{13}
$$

where $N_{RT_i}$ is the number of times that the CSMA/CA algorithm goes back to calculate a delay from the time the channel was found to be busy. This is a random variable. That is, it takes a different value for each transmission. The $(P_{TX_i} + P_{RTX_i} + N_{RT_i})$ term implies that before each message transmission, the channel auditing process must be executed (CSMA/CA algorithm). In the best case, $P_{TX_i} = 1$ in the $(P_{TX_i} + P_{RTX_i} + N_{RT_i})$ term, because the channel was free. Then $N_{RT_i} = 0$, (there were no retries listening to the channel, i.e., CSMA retries), and the packet is transmitted successfully with $P_{RTX_i} = 0$ (without retransmissions). However, the network conditions are not always ideal and there will be collisions causing packet retransmissions and retries when listening to the channel. Then, the $N_{RT_i}$ and $P_{RTX_i}$ variables will exhibit non-zero values.

Throughout the sampling time ($T_{Sampling}$), the node consumes energy from the microcontroller in active mode for this model. Therefore,

$$
E_{MC_i} = T_{Sampling} \times I_{MC} \times V_{MC}.
\tag{14}
$$

Now, when the sampling time begins, all nodes turn on. Then, the initial energy consumed by the node *i* is $E_{ON_i} = T_{ON} \times I_{ON} \times V_{ON}$. Finally, when the time has finished, node *i* turns off, consuming shutdown energy, so $E_{OFF_i} = (T_{OFF} \times I_{OFF} \times V_{OFF})$.

At the end of the sampling period, the node reports the total energy consumed by all the functions it performed during the network processes, which will be the local energy of the node. Thus,

$$
\begin{aligned}
EnergyNode_i = \quad & E_{ON_i} + E_{MC_i} + E_{OFF_i} + \\
& E_{Switching_i} + E_{CSMA_i} + E_{TX_i} + E_{RX_i}.
\end{aligned}
\tag{15}
$$

To obtain the total node energy to assess the global energy of the network, we add the total energy of each node when the sampling time has ended as shown in (16):

$$
Total_E nergy = \sum_{i=1}^{Total_Nodes} EnergyNode_i,
\tag{16}
$$

where *totalNodes* is the total number of nodes.

Equation (15) describes the actions performed by node *i* since it turns on (consuming initial energy $E_{ON_i}$), and becomes part of a network, until the moment it turns off. The most complex stage is when the node listens to the channel. This activity causes the node to decide whether to transmit packets. All packets sent (either transmitted or retransmitted) imply that the node must first listen to the channel. That is, the node executed the CSMA/CA algorithm using CSMA energy ($E_{CSMA_i}$). If the channel is free, the node consumes transmission energy ($E_{TX_i}$) to transmit the packet ($P_{TX_i}$). If the channel is busy, the node generates a retry variable ($N_{RT_i}$) because it has to execute the CSMA/CA algorithm again. After the node sends the packet, it receives an acknowledgment, ACK, using reception energy ($E_{RX_i}$) and receives the packet ($P_{RX_i}$). On the contrary, if the node does not receive an ACK, it retransmits the packet generated as a retransmission ($P_{RTX_i}$). For all activity changes, the node consumes switching energy ($E_{Switching_i}$). After this basic single packet cycle, the node continues to receive, listen, transmit,

and changes states until the simulation time has ended and the node turns off using shutdown energy ($E_{OFF_i}$). During all the processes performed by a node as part of a network, the node is in active mode and the microcontroller remains active throughout the sampling period, consuming microcontroller energy ($E_{MC_i}$).

Algorithm 1 shows the model operation pseudo-code for each of the energies previously exposed in the main processes executed by a node within the network.

---

**Algorithm 1:** Algorithm for the proposed energy model.

```
E_{ON–Total} = 0
E_{MC–Total} = 0
E_{OFF–Total} = 0
E_{Switching–Total} = 0
E_{CSMA/CA–Total} = 0
E_{TX–Total} = 0
E_{RX–Total} = 0
FOR{Every node i}
Node i turns on with a coverage radius;
E_{ON–Total} = E_{ON–Total} + E_{ON_i}
E_{MC–Total} = E_{MC–Total} + E_{MC_i}
Node i listens to the channel;
E_{CSMA/CA–Total} = E_{CSMA/CA–Total} + E_{CSMA/CA_i}
E_{MC–Total} = E_{MC–Total} + E_{MC_i}
IF{channel is free }
Send neighbor acknowledgment packet;
E_{TX–Total} = E_{TX–Total} + E_{TX_i}
E_{RX–Total} = E_{RX–Total} + E_{RX_i}
E_{MC–Total} = E_{MC–Total} + E_{MC_i}
ELSE
Increase delay and re−check channel;
E_{MC–Total} = E_{MC–Total} + E_{MC_i}
END IF
Node i populates its neighbors table;
IF{Node i requires sending information}
Find a route in neighbors table;
IF{channel is free}
Send neighbor acknowledgment packet;
E_{TX–Total} = E_{TX–Total} + E_{TX_i}
E_{RX–Total} = E_{RX–Total} + E_{RX_i}
E_{MC–Total} = E_{MC–Total} + E_{MC_i}
ELSE
Increase delay and re−check channel;
E_{MC–Total} = E_{MC–Total} + E_{MC_i}
END IF
END IF
IF{Simulation Time == end}
E_{OFF–Total} = E_{OFF–Total} + E_{OFF_i}
E_{MC–Total} = E_{MC–Total} + E_{MC_i}
END IF
END FOR
```

---

*2.1. Limitations of the Energy Model*

This subsection describes some of the limitations of the proposed energy model. This model comprises the main types of energy required for a node to perform generic functions within the network. However, this model does not describe specific energies related to the anomalous behavior of the nodes, yet it is capable of predicting strange behaviors in the network due to the alteration of typical node functions. These behaviors are due to situations of stress, such as node connections and disconnections due to channel intermittence. Likewise, these situations can be represented as a result of other types of energy. For example, in network areas with intermittent nodes or links, nodes frequently disconnect. This situation reflects in increased retransmissions, channel auditing retries, and packet collisions. These changes increase the transmission, reception, switching, and energies of the CSMA/CA algorithm. Thus, the proposed model can identify anomalous behavior observed in these types of energy and draw conclusions regarding risk areas or potential attacks.

The main function of a WSN is sending information to one or more collector nodes and detecting the conditions of the channel to find the optimal route. Then, in these networks, it is possible to implement different sleeping node techniques. That is to say, there are periods of time in which a certain number of nodes enter into passive mode, consuming less energy than they normally do. In this study, we do not look into sleeping node energy. However, the model can be extended, adding this energy equation, if the voltage and current of a sensor in a given time interval are available. For these reasons, the proposed model is simple, easy to implement, and quite scalable as per analysis requirements.

## 3. Analysis and Results under the Simulation Tool

One of the objectives of this work is to assess critical metrics in WSNs. In this section, the performance of the MPH routing protocol is compared against three sensor network protocols: AODV, DSR, and ZTR. The performance metrics used for comparison are the total energy consumption, delay, overload, resilience, valid routes, and number of jumps. This analysis is performed to analyze the influence of these parameters on energy consumption.

This is used to assess possible anomalies in a network or an area at risk of attack. The delay and energy consumption in a network are directly related to the complexity of the routing algorithms. When a routing protocol uses many algorithms and processes to send a packet, the sensor nodes will introduce long delays and incur in high-energy consumption. The MPH routing protocol is not only characterized by its hierarchical topology, but also by the origin touting and fast reconfiguration of its topology when faced to an unexpected change. This has to do with the resilience of the network (the network's ability to recover from unexpected changes).

A routing protocol provides reliability when nodes exhibit in their tables (either routing or neighbor tables) valid routes, i.e., routes that have not expired or that have not been invalidated by a node disconnection. Overloading the routing protocol will affect the channel's occupation because it is constituted by the control packets that a routing protocol needs for the reliable delivery of information. The lower the overload, the lower the probability of collisions and, in turn, the lower the number of packet retransmissions. A small overhead ensures that the information will be delivered more quickly and reliably.

For simulations, a grid layout with 22 nodes described in Figure 3 is considered. The nodes are located randomly in an area of 300 m × 300 m and nodes have a coverage range of 15 m. A data rate of 250 kbps and a packet length of 22 bytes were taken. Table 3 shows the simulation parameters. Each node contributes a traffic of 10 packets per second. There are two coordinator nodes marked in red color. All network nodes can send packets to the coordinator nodes directly (one hop) or indirectly (multiple hops that form a route). There are nodes near to the coordinator and all other nodes must send packets through these nodes, causing them to drain their batteries faster because they constitute the network bottlenecks.

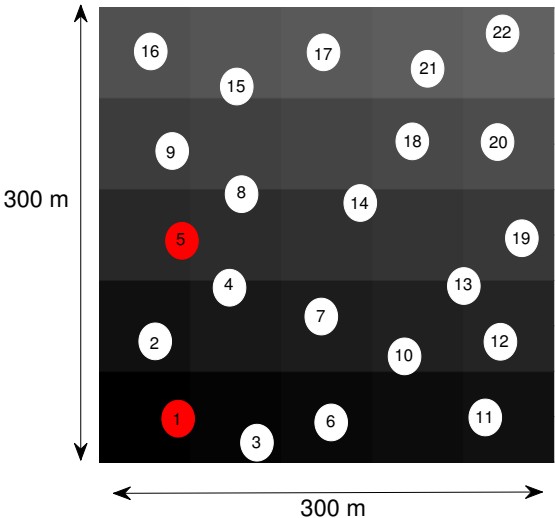

**Figure 3.** The network topology considered.

**Table 3.** Simulation and real network parameters under CSMA/CA carrier sense multiple access with collision avoidance [34].

| Parameter | Value |
|---|---|
| **Physical Layer Parameters** | |
| Sensitivity threshold receiver | −94 dBm |
| Transmission power | 4.5 dBm |
| **MAC Layer Parameters** | |
| Maximum retransmission number | 3 |
| Maximum retry number | 5 |
| Maximum number of tries to reach a node from the collector | 9 |
| Packet error rate | 1% |
| Average frame length | 22 bytes |
| Maximum number of backoffs | 4 |
| MAC protocol | IEEE 802.15.4 |
| MAC layer | CSMA/CA |
| **Network Layer Parameters** | |
| Number of nodes | 22 |
| Maximum data rate | 250 kbps |
| Scenario | Static nodes |

Figures 4–9 show, for some well-known protocols in the literature, the energy types per crown for the distribution shown in Figure 3. These graphs show the numerical analysis for each type of energy described in the proposed energy model. A 24 h/7 day execution of 22 sensors was carried out in an area of 300 × 300 m$^2$ of the university campus of Universidad Panamericana in Guadalajara, Mexico. A set of high-level wireless communication nodes CC2650 and CC2530, based on the IEEE 802.15.4 standard, were used. The space covered by the sensors is an area with trees, buildings, traffic of people and vehicles with both indoor and outdoor spaces. The distribution structure of the sensors is the one shown in Figure 3. We aimed at taking six representative protocols in WSN such as: AODV, DSR, LEACH, PEGASIS, MPH (proposed protocol), and ZTR. Crowns were formed by the proximity of the nodes with respect to the coordinating node. A crown has the characteristic that the nodes belonging to it have similarity with respect to their performance parameters, i.e., with approximately the same distance to the coordinating node, the nodes have more or less the same traffic and forwarding packets

to their destiny. This is why the nodes are grouped into categories called crowns, which make them take a specific level of similarity in the network. For these graph results, the network will only have a coordinating node and this will be node 1. It is important to note that crowns are formed based on the amount of links that each routing protocol forms; not all protocols have the same number of crowns to analyze. The proposed energy model lets us know exactly how much energy an average node spends on each crown. The energy model is applied to the programming and operation of the sensors to establish their energy consumption separately, depending on the type of energy. As the model is validated, it allows global and local knowledge of the energy of the nodes according to the performance parameters such as: the proximity to the coordinating node, the number of established links, the collisions that generate the amount of traffic and control packets, the delay in the delivery of information, and the processing of resources at the node level.

In Figure 4, the AODV protocol presents six crowns of nodes. This may be because AODV is a reactive protocol that forms links in all directions, so the network becomes a mesh with widely redundant links. However, the fact of having so many links may produce an increase in packet collisions and protocol control packets, which generates more packet retransmissions and listening attempts to the communication channel. The advantage of presenting several crowns or levels of nodes with similar performance is the redundancy and the amount of different routes that a package can take to reach its destination. The disadvantage presented by this number of packets administered by the routing protocol is the high-energy consumption due to the possible loss of packets and their retransmission. In AODV, the crowns with the highest energy load are 1 and 2, which are the closest nodes to the coordinating node. This is understandable because they are the nodes that forward the traffic of the other nodes in the network and the coordinating node generates a bottleneck. We observe that between the last crown (crown 6) and the first two crowns, the difference in energy expenditure is 48%, which shows that nodes farther from the coordinating node have less traffic load, less collisions and less retries listening to the channel, then the CSMA algorithm runs in less times. Due to the energy model, we can also note that in crowns 1, 2, 3 and 4 the transmission and CSMA energies are similar. This may be due to the strong weight of the network that is located at the center of the topology; the AODV protocol creates a mesh and not a link tree.

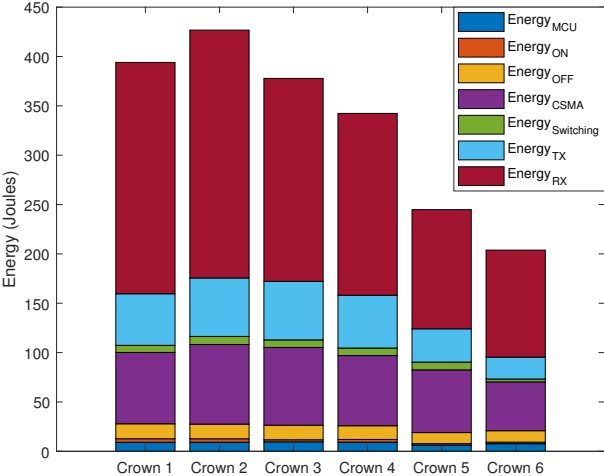

**Figure 4.** Energy types per crown for the AODV protocol.

In Figure 5, the DSR protocol presents a structure and a quantity of crowns similar to AODV. These two protocols are the highest energy consumers of all the protocols studied in this work, with a 15% higher expenditure of energy in the nodes. Crowns 1 and 2 have the highest energy expenditure due to the redundancy of the protocol links. Although in DSR the links form a mesh in the network and there is a large amount of packet flow, this protocol, unlike AODV, has a more marked energy

expenditure per crown; the first being the most consumed, and the last, the one that consume the less. For DSR, the energy difference between the first and the last crown is 42%. In addition, due to the topology configuration that DSR generates, there are many surrounding packets in the network (traffic and control), which generate packet losses, retransmissions and therefore, listening retries to the communications channel to determine if it is already available or still in use. This can be noted with the fact that CSMA energy is similar in almost all crowns. The AODV and DSR protocols, being both reagents, have a similar energy expenditure with a difference of only 6%, even though in the last crown, the transmission energy decreases in DSR because the crowns are more scaled than in AODV.

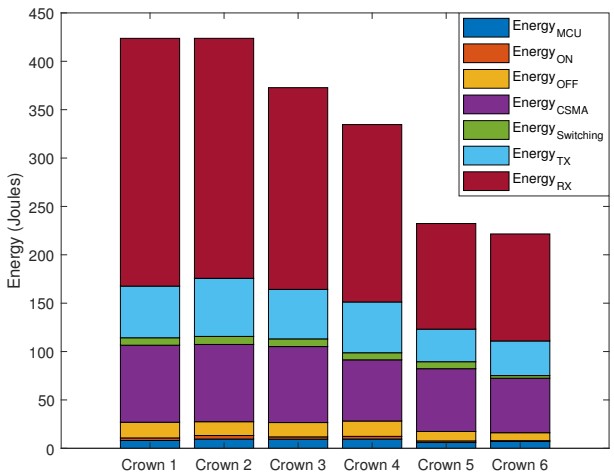

**Figure 5.** Energy types per crown for the DSR protocol.

In Figure 6, the LEACH protocol uses techniques to reduce collisions between clusters and within the clusters themselves. Data collection is centralized and runs periodically, which is a characteristic of a proactive protocol. The protocol configuration for this scenario points that for one third of each day, a node near the lower left corner of the topology (near node 1) will be the coordinating node, for another third of the day, it will be a node in the middle of the topology, and finally, for the remaining third, it will be a node near the top right corner of the topology (near node 22). Due to the imbalance that is established in these changes of roles of the nodes, there is less crowns in the network and almost all have approximately the same energy expenditure. The above allows stating that all nodes belong to the same crown. LEACH assumes that all nodes transmit with sufficient power to reach the coordinating node and that each node has sufficient computing power to support different MAC protocols. In practice, this is complicated and, as it can be seen in real cases, the first crown differs energetically from the last one by 6%. However, the existence of cluster and various roles of the nodes, allows reducing energy by 13% with respect to reactive protocols such as AODV and DSR. In LEACH, the transmission energy is almost the same in all crowns, except for the last one, with a difference of 5% with respect to the others.

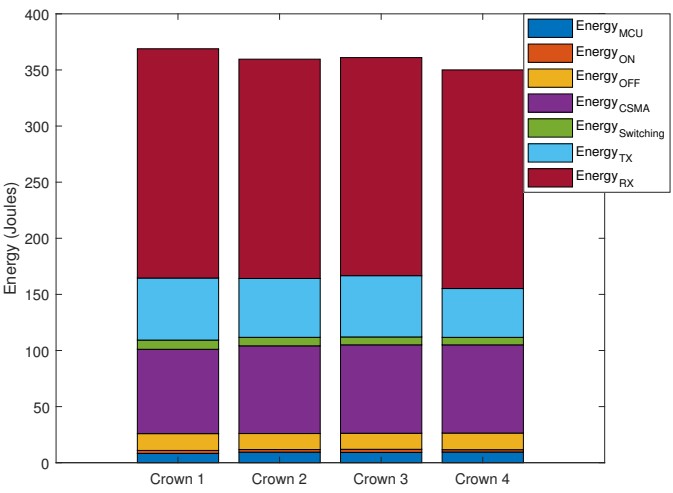

**Figure 6.** Energy types per crown for the LEACH protocol.

In Figure 7, the PEGASIS protocol extends the life of the network by limiting the nodes communication only to their closest neighbors and take turns communicating with the coordinating node. This protocol uses networking techniques and allows only local traffic between nodes that belong to the same region or crown to reduce bandwidth consumption. One of the great advantages of PEGASIS is that the distance between the nodes is calculated based on the intensity of the signal; the links are really strong, thus preventing packet retransmissions. We observe that the difference in energy consumption between LEACH and PEGASIS is 14%, with PEGASIS demonstrating the greatest savings. Energy improvement occurs by avoiding overload caused by LEACH's dynamic generation of the cluster and by minimizing the number of packet transmissions and receptions using the data aggregation technique. PEGASIS assumes that each node must be able to communicate with the coordinating node directly and that each node contains a complete database of the location of the other nodes in the network. This reduces network performance a bit by making processing slightly heavier. The energy of the crowns is similar, only with a difference between them of 5% and due to the role relay in the nodes; the crowns are not scaled from higher to lower consumption.

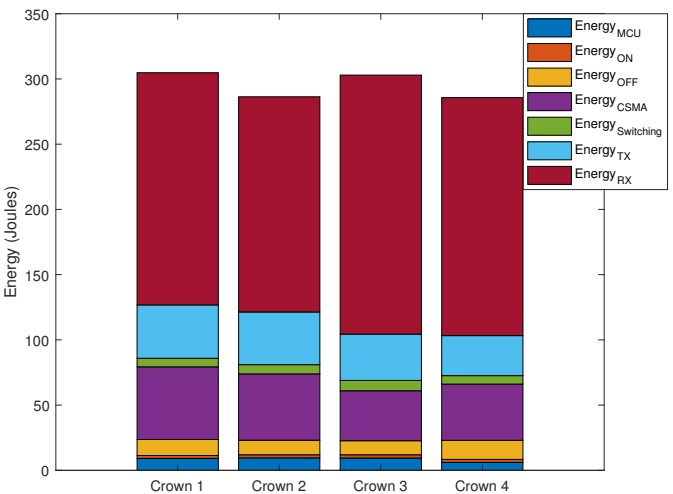

**Figure 7.** Energy types per crown for the PEGASIS protocol.

In Figure 8, the MPH protocol is a hybrid protocol (predominantly proactive). It establishes link hierarchies based on the proximity of a node to the coordinating node. This hierarchical tree topology only allows a few links, but there is a sufficient degree of redundancy. In MPH, we observe five crowns scaled around the coordinating node with an energy difference of 10% between the first and the last one with a similar transmission/reception energies in all crowns. MPH takes advantage of the fact that there are no links among nodes of the same hierarchy level, which decreases the cost in processing the neighbor tables and decreases the amount of protocol control packets. The difference in energy consumption between AODV and DSR with respect to MPH is 40% in favor of MPH.

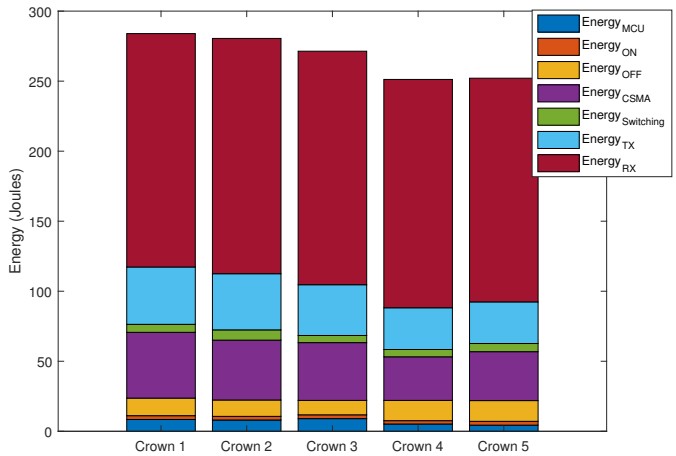

**Figure 8.** Energy types per crown for the MPH protocol.

The ZTR protocol, shown in Figure 9, is a proactive protocol, simple, and easy to implement. It consists of an algorithm with limited resources that performs multi-hop routing without route discovery procedures and is based on a hierarchical distribution scheme. As in MPH, we note that five scaled crowns are established from the highest to the lowest energy consumption. ZTR has a 5% energy saving with respect to MPH because its links are simpler and the nodes cannot have more than one parent node. The small difference in energy expenditure that is established between MPH and ZTR, being ZTR so simple, is because several packages can be lost due to the low redundancy of links but this fact is compensated with the ZTR's speed of information delivery.

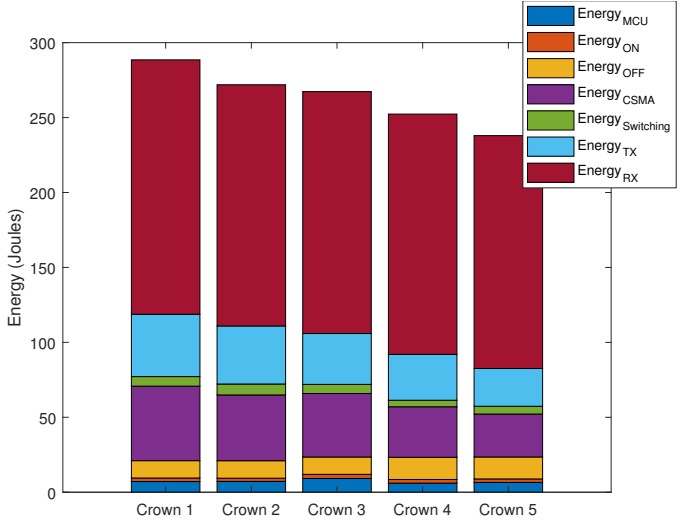

**Figure 9.** Energy types per crown for the ZTR protocol.

Now, having tested the energy model for the random scenario, Figure 10 shows the energy distribution of the nodes for each observed protocol: AODV, DSR, LEACH, PEGASIS, MPH, and ZTR. This scenario is based on the topology presented in Figure 3. The packet sending rate is 10 packets per second and the links have a packet loss between 0.5 and 1.5%, measured on a one day window with an average per hour for each node. According to the results, it can be observed that approximately 75% of the nodes in MPH and ZTR present values below 0.10 Joules while for AODV, 75% of the nodes distribute their energy in values between 0.14 and 0.19 Joules. Regarding DSR, the energy values for the 75% of the nodes are distributed between 0.16 and 0.19 Joules. The MPH and ZTR protocols present some extreme values that show their proactive nature, in which the creation of hierarchical routes and the amount of energy can be concentrated in the nodes near to the collector nodes. We can also note that the MPH protocol has a more compact energy distribution between randomly distributed nodes, even more compact than ZTR since both have lower energy consumption compared to AODV and DSR. Compared to DSR, AODV has a marked distribution of energy between nodes; this indicates that the crowns, which nodes forward packets and are around the coordinator nodes, have more significant differences in energy consumption in AODV than in DSR. LEACH and PEGASIS present the most compact distribution of energy in the network nodes. The values of 50% of the nodes range between 0.06 and 0.078 Joules for LEACH and their most extreme value falls to 0.042 Joules, this can occur in the crowns furthest from the coordinating node. In this protocol, the farthest nodes from the coordinating node can lead to a shorter survival time and generate greater packet delays. For PEGASIS, 50% of the nodes have energy values between 0.06 and 0.07 Joules and have an outlier at 0.041 Joules. Due to the propagation of chain packets, this protocol is the most efficient in energy consumption. This protocol reduces both the bandwidth requirement and the overhead.

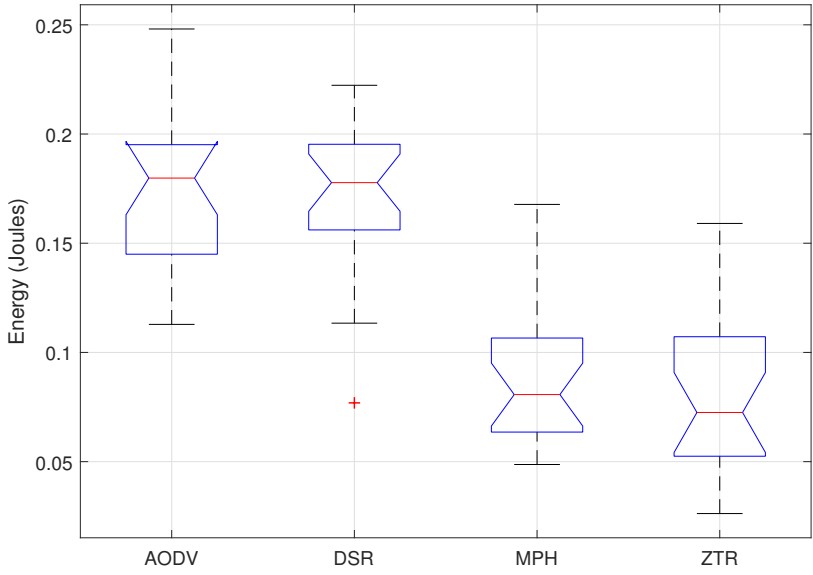

**Figure 10.** Energy distribution in the nodes for each evaluated protocol.

In Figure 11, energy transmission is varied in order to increase the coverage radius. Its aim is to show how energy behaves in each of the studied protocols. These conditions can also evaluate some type of resilience in the network according to each protocol. The fact of increasing the coverage radius generates more collisions in the network. Although the routes can be compensated, because they are shorter, nodes have a greater number of directly connected neighbors. Even so, the energy model shows that the MPH protocol has similar characteristics to ZTR, the latter being extremely simple and not scalable but quite fast and efficient. MPH shows an average energy saving of 10% with respect to ZTR, 24% with respect to DSR, and 28% compared to AODV.

In addition, we observe that LEACH and PEGASIS have similar behaviors with a difference of 3% between them. These are the protocols that have the lower energy consumption in face of the stress caused in the network. When the transmission power is increased, these protocols maintain the rotation of roles in the nodes and therefore, both the energy and the network overload are balanced. The above is perfectly combined with the fact that both are hierarchical protocols further enabling packet traffic. PEGASIS presents a decrease in the total energy for each radio due to its approach of sending packets in chain, unlike the establishment of clusters exhibited in LEACH. MPH differs from LEACH and PEGASIS by 30%. This may be due to the proactive nature of MPH, in which from time to time, the tables of neighbors are updated and generate greater overhead. If the radius increases, the tables of neighbors become bigger and their update more complex.

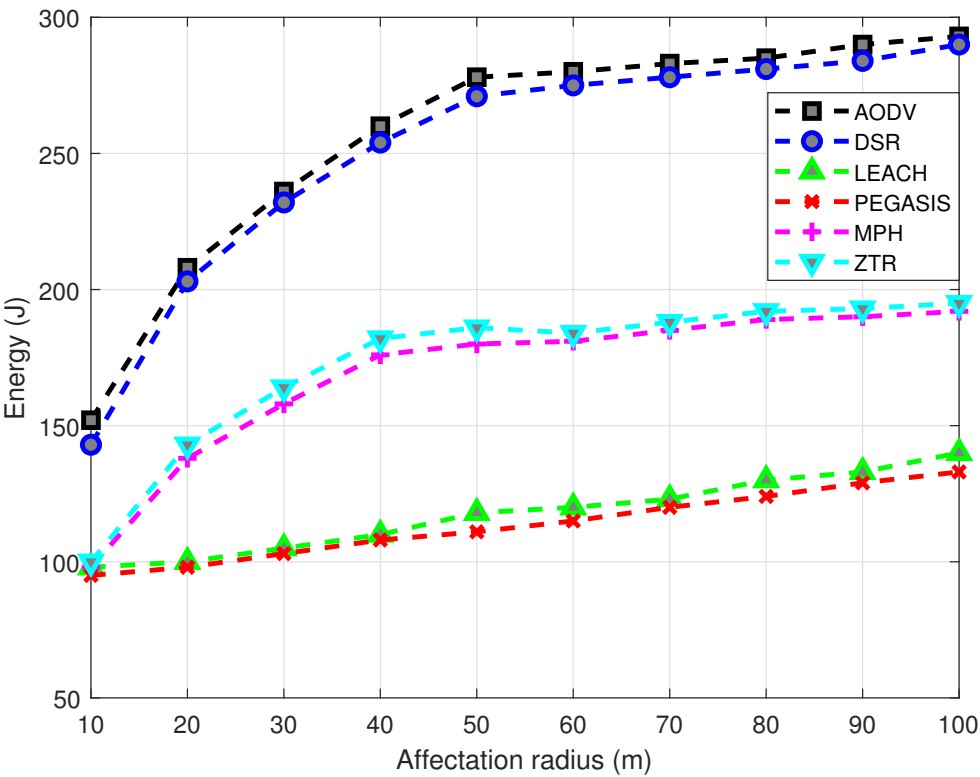

**Figure 11.** Energy according to the increase in the coverage radius of a node.

Figure 12 shows two comparative conditions for the six protocols under study: stable network conditions working properly and adverse network conditions under the topology referred in Figure 3. For the study, we took an average of energy at the same hour (noon) for 7 days for each of the nodes in the network. Nodes transmit at a rate of 70 packets per second. To generate interference and create adverse conditions in the network, we put 5 nodes close to the majority of nodes in the network that were emitting the same reactive jamming frequency, thus increasing the loss of packets along the links as it is shown in Figure 13 with unnumbered jammer nodes marked in blue. The network has a stress zone and a high focus on packet loss. This was made to analyze how the network reacts with each routing protocol and how this influences the energy of each node having both local and global perspectives regarding energy consumption of the network.

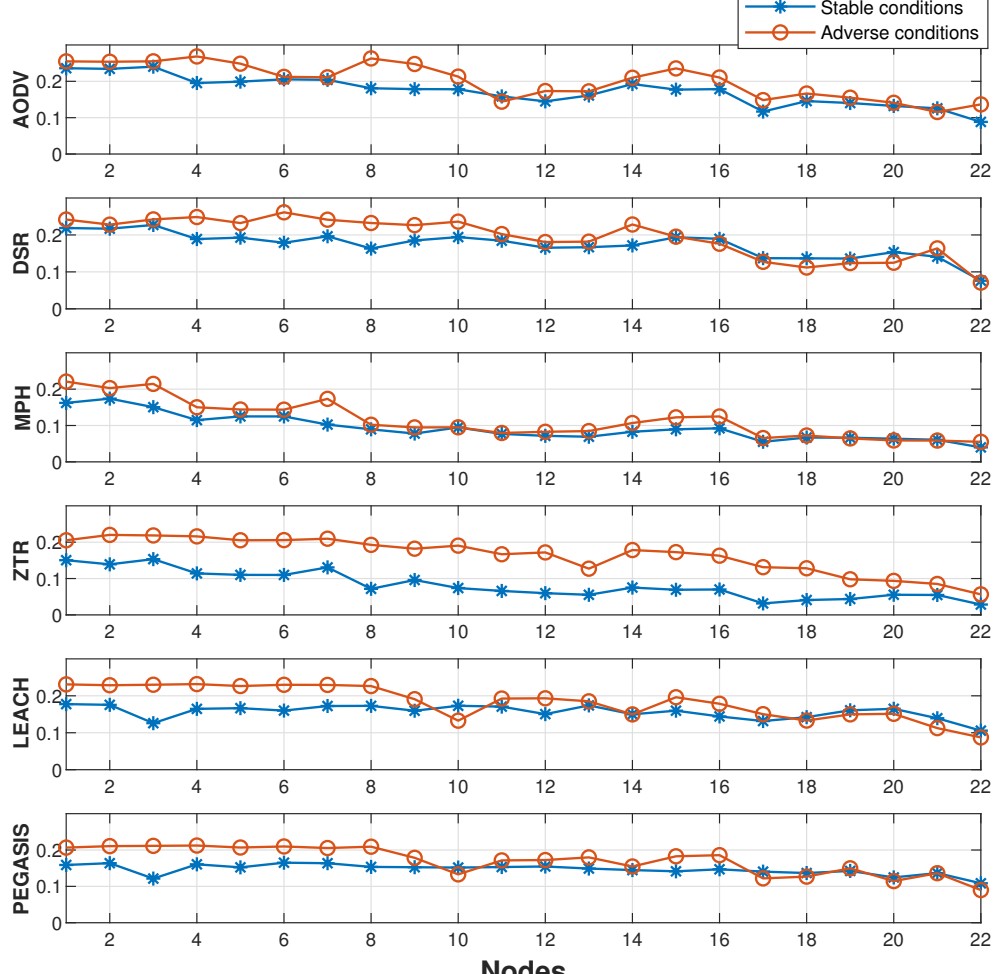

**Figure 12.** Energy at each node of the network under both stable and adverse conditions for the four protocols studied.

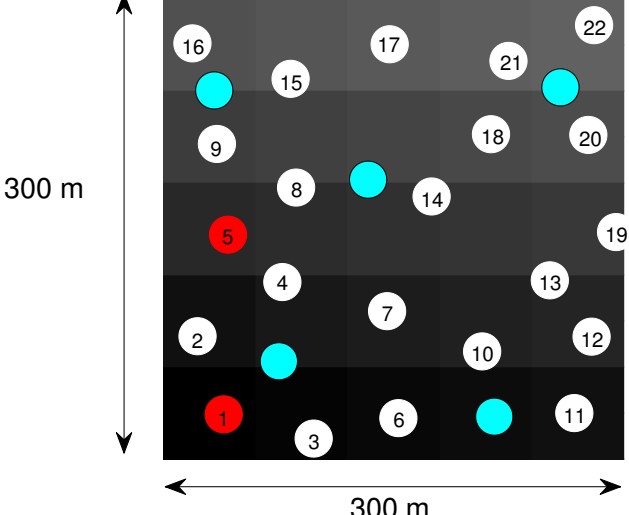

**Figure 13.** Grid topology with jammer nodes generating adverse conditions.

We observe that node energy in the surroundings of that half greatly increased in adverse conditions for AODV and DSR. ZTR and MPH try to stabilize the energy of the network nodes and redistribute the packet losses. LEACH and PEGASIS have a similar energetic behavior that does not necessarily behave by crowns from greater to lesser overload. These protocols react very well before adverse conditions because in more than 50% of the nodes, the energy expenditure is similar in both stable and adverse states. These protocols have a similar energy expenditure between them with a difference of 5% in favor of PEGASIS and a difference of only 3% with respect to MPH and ZTR. This is reflected in the fact that the average energy per node in adverse conditions for the network are 0.1990, 0.1968, 0.1805, 0.1741, 0.1671, and 0.1150 Joules for AODV, DSR, LEACH, PEGASIS, ZTR, and MPH, respectively. The above figures show that MPH has an energy expenditure similar to ZTR, which is a simple and fast algorithm; still MPH exhibits the advantage of route maintenance and redundancy. ZTR and MPH have similar natures, both having proactive characteristics. However, when there is a large number of packets in the network, there are more collisions and packet loss. The ZTR protocol can lose links and some isolated nodes might remain causing an increase in energy consumption. For this reason, the energy difference between MPH and ZTR is 31%. MPH, having multi-parent links, shows greater redundancy and increases the amount of valid routes that continue to function for changes in the network. This can be interpreted taking into account that MPH was designed to combine the best design features of a proactive protocol with the redundancy of the reactive protocols and the foregoing together with the verification of the proposed energy model, which allows visualizing easily and with concise data possible anomalies in specific areas of a network.

In Table 4 a sampling period of 100 seconds was taken for the simulations. Tests were performed every 10 seconds, which is the percentage of Packet Delivery Ratio (PDR). The PDR is a metric that indicates the number of packages delivered in a given time and indicates the collisions present in a network that cause packet loss. This performance metric is directly related to the energy and provides an idea on how nodes behave in the network; for example, if there are connections and disconnections of nodes, interference and quality of the links, among other features that show the performance of the routing protocol. As aforementioned, AODV and DSR have a similar behavior because their reactive nature. Initially, the network nodes do not know the routes to the destination, so they send request packets implying a large number of control packages, which increases the overhead. The LEACH and PEGASIS protocols try to make the route to the coordinating node efficiently through clusters or changes in the role of the coordinating nodes; this increases the assertiveness of the nodes in the package delivery, a consequence that is reflected in energy consumption. The difference in PDR in this initial stage between AODV and DSR with respect to LEACH and PEGASIS is approximately 23% in favor of the latter protocols. MPH and ZTR are proactive protocols, so at the initial stage, nodes behave the same as in stable state because the neighbor tables are periodically renewed and not when the routes require it. One of the big differences between MPH and ZTR is multi-parent links, which allow the packets to have greater redundancy in MPH and a reliable delivery. The difference in PDR between AODV and DSR with respect to MPH and ZTR is 28% in favor of MPH and ZTR.

**Table 4.** Percentage of Packet Delivery Ratio (PDR).

| TIME (s) | | | % | PDR | | |
|---|---|---|---|---|---|---|
| | AODV | DSR | LEACH | PEGASIS | MPH | ZTR |
| 10 | 72 | 73 | 92 | 93 | 98 | 93 |
| 20 | 81 | 84 | 90 | 93 | 98 | 89 |
| 30 | 92 | 91 | 91 | 94 | 98 | 89 |
| 40 | 97 | 97 | 91 | 93 | 98 | 90 |
| 50 | 93 | 91 | 92 | 94 | 96 | 91 |
| 60 | 83 | 84 | 93 | 95 | 97 | 91 |
| 70 | 86 | 86 | 93 | 93 | 99 | 91 |
| 80 | 93 | 93 | 90 | 93 | 98 | 90 |
| 90 | 96 | 97 | 90 | 93 | 96 | 89 |
| 100 | 92 | 93 | 90 | 94 | 98 | 91 |

Concerning the rate of transmitted packets, we analyze the measurement impact of the energy model when we vary the packet transmission rate. Tests were carried out for 1 day (24 h). In particular, on a Wednesday; the day in which we can find regular traffic of people and vehicles on campus, and therefore, more operating wireless devices. We show these results in Table 5.

**Table 5.** Effect of packet transmission rate (PTR) variation on energy.

| PTR (kbps) | | | Total | Energy (J) | | |
|---|---|---|---|---|---|---|
| | AODV | DSR | LEACH | PEGASIS | MPH | ZTR |
| 50 | 33.4 | 31.3 | 20.4 | 20.1 | 21.1 | 20.2 |
| 100 | 35.4 | 34.7 | 23.8 | 24.3 | 25.4 | 23.1 |
| 150 | 42.3 | 41.5 | 30.4 | 27.4 | 32.5 | 31.2 |
| 200 | 44.6 | 43.2 | 32.1 | 30.2 | 34.2 | 34.5 |
| 250 | 45.8 | 43.1 | 32.3 | 31.4 | 34.6 | 35.1 |

We note that, for five different packet transmission rates, the most drastic variation is found in the rates of 50 and 100 kbps, with a 37% difference with respect to energy at 250 kpbs. The difference in energy expenditure with the change in packet transmission rates is 43% higher for reactive protocols, compared to other protocols. The difference between proactive protocols and energy-aware protocols is only 5% in favor of the latter.

*3.1. Experimental Scenario*

To validate the tests done by the simulator, we intended to recreate a scenario with real sensors on the university campus of the Universidad Panamericana in Guadalajara, Mexico. We considered an area of $300 \times 300$ m$^2$ surrounded by a two-story building, a green area, living rooms, and free area with a terrace, as shown in Figure 14.

Figure 15 shows the top view of the engineering building at the university campus and the location of the sensors in this building. This plan helps to better visualize the walls and the specifications of the construction to give a better idea of the traffic routes of people, location of computer equipment, lights, etc.

There is a Gateway with LoRa technology and WiFi, which is the general hub of all the nodes. This Gateway receives data from all devices and sends the information to a platform called the things network. The devices that are held as nodes are divided into three technologies: Bluetooth, LoRa, and Zigbee.

Zigbee nodes are a set of high-level wireless communication protocols, based on the IEEE 802.15.4 standard, i.e., communication using the 2.4 GHz frequency. The CC2530 device is configured as the coordinator; its main function is to create the network in mesh topology. The other four devices can be connected to the network and have communication between them, allowing the possibility to have devices withdrawn from the coordinator because the information packet passes through the other

devices to the coordinator. Each of the devices contains a temperature sensor, allowing measuring the ambient temperature in different locations of the university campus. The coordinator sends a signal to know which devices are connected to the network and in turn, these send their temperatures every 5 min. After obtaining all data, the coordinator sends the information through another protocol called LoRa to the Gateway.

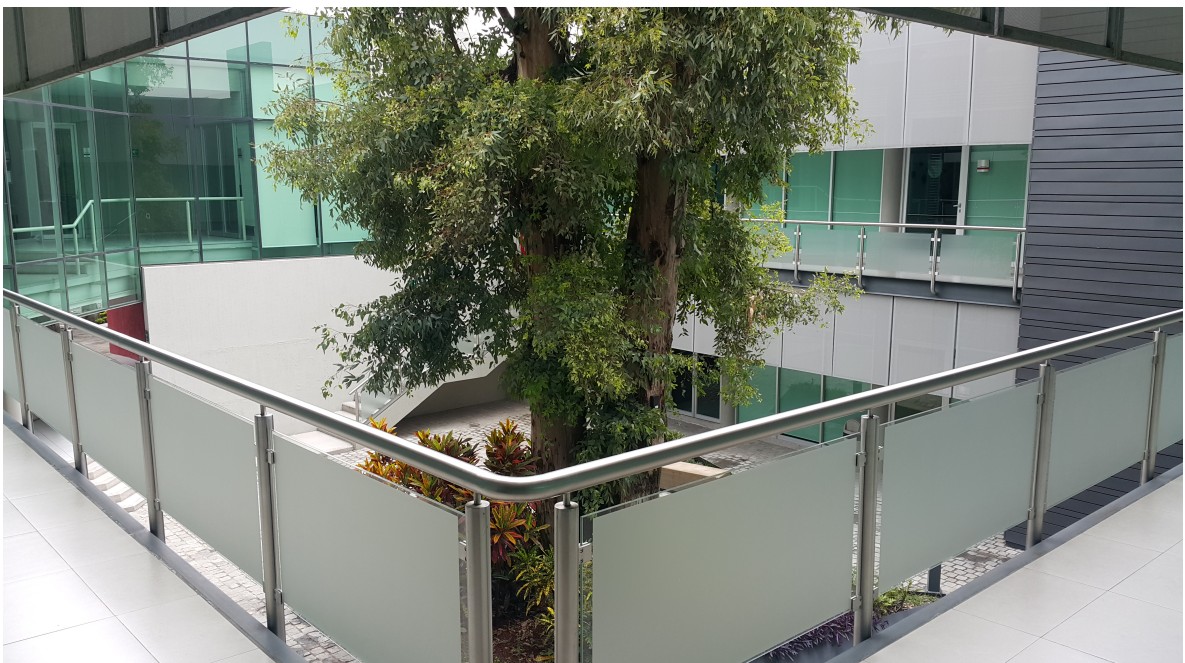

**Figure 14.** The test area at Universidad Panamericana (Mexico).

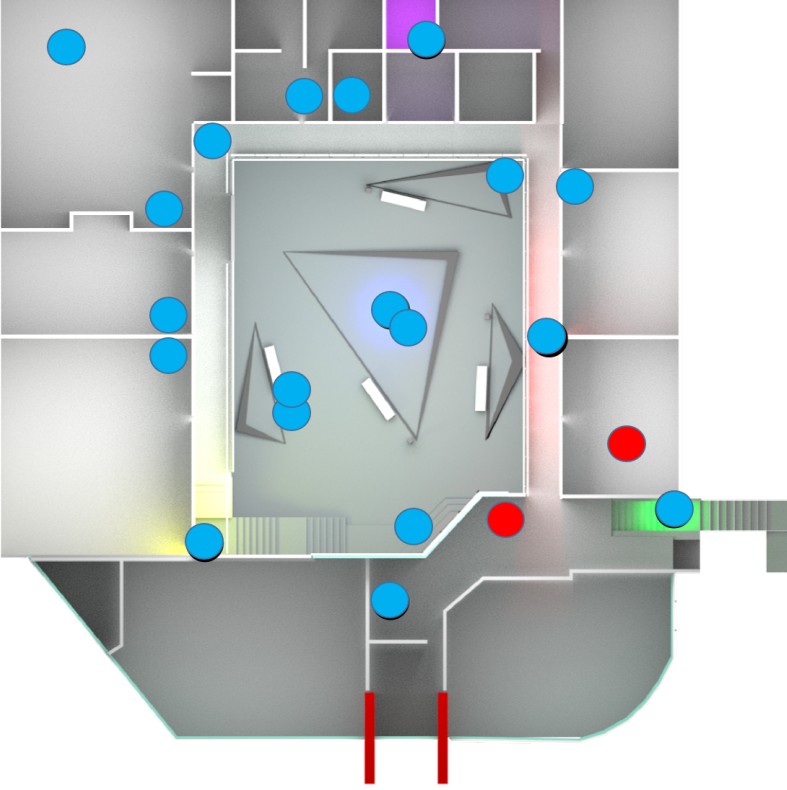

**Figure 15.** Top view of the test area (engineering building at Universidad Panamericana, Mexico).

For LoRa nodes, LPWAN is a specification for low power and wide area networks designed specifically for low power consumption devices operating in local, regional, national, or global networks. In Mexico, LoRa uses the 915 MHz frequency. The topology of LoRa is point-to-point; there is a gateway or hub and one or more nodes. The Gateway is in charge of reading all the packets that are on that frequency. The nodes are devices that transmit small information frames to avoid high-energy consumption.

The created network consists of five different devices: garbage sensor, light sensor, accelerometer sensor, gyroscope sensor, and environmental sensor. The Gateway is the device that acts as connection interface between devices and allows resource sharing between two or more computers. The Gateway used contains the LoRa and WiFi protocols. It obtains all the transmitted data through the LoRa protocol; data are then transmitted via WiFi allowing them to be found on the ThethingsNetwork platform. In this platform, it is possible to display the data separately from each node, knowing when the last transmission was made, the frequency of transmission of each device, and its measurements. This device obtains all the data transmitted by the Zigbee coordinator, all the data transmitted by the LoRa nodes, and also the data transmitted by the Bluetooth concentrator.

For Bluetooth nodes, Wireless Personal Area Networks (WPAN) is an industrial specification operating on the frequency of 2.4 GHz as well as Zigbee. There are Bluetooth nodes that work as beacons (low power consumption devices that emit a broadcast signal). In this case, each node has a light sensor and they are constantly sending light values. In addition, there is a fifth device, which works as beacon scanner. The function of this device is to receive all the data coming from the other nodes, decode them, and send them through LoRa using another module that contains that protocol.

Figure 16 describes the devices used for the real scenario according to the wireless technologies mentioned above. A communications sniffer was also used in addition to the application interface www.thethingsnetwork.org. In the middle of the figure, two devices act as coordinating nodes. The distribution of the nodes in real space is shown in Figure 17 where we have an area of approximately 300 m × 300 m that is displayed in Figure 14. We show an approximate radius of coverage of the sensors of 40 m. However, it can vary according to the antenna. Empirically it can be smaller by the amount of collisions according to wireless technology.

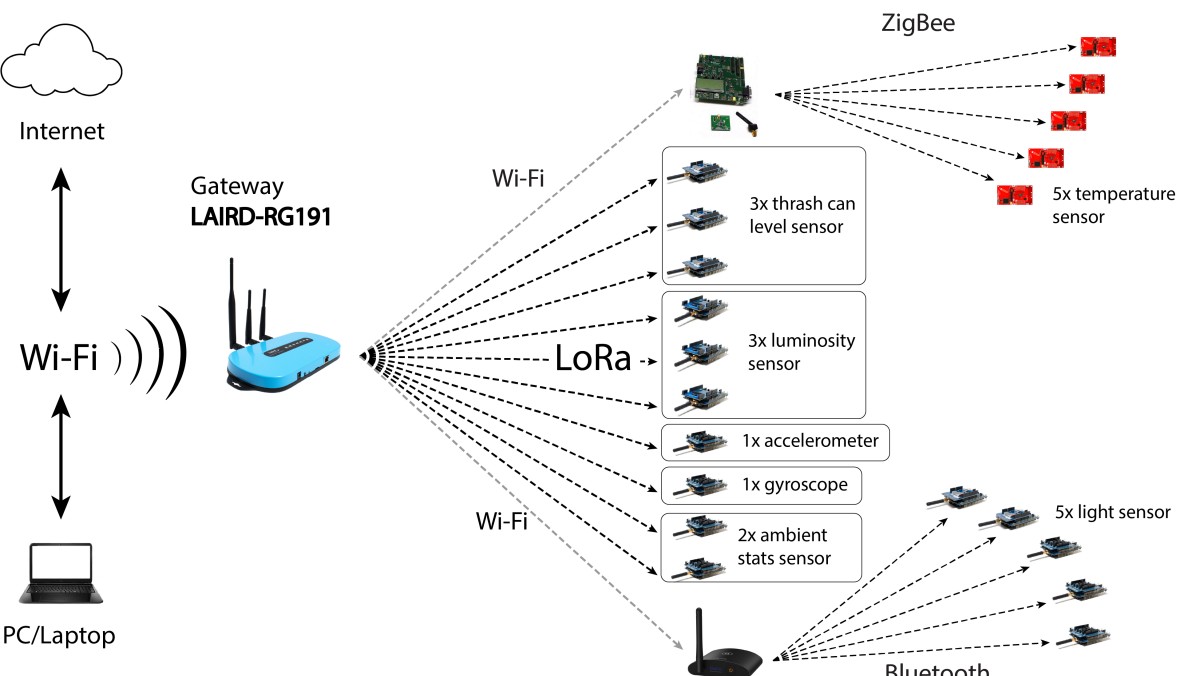

**Figure 16.** Devices used for the experimental validation [40].

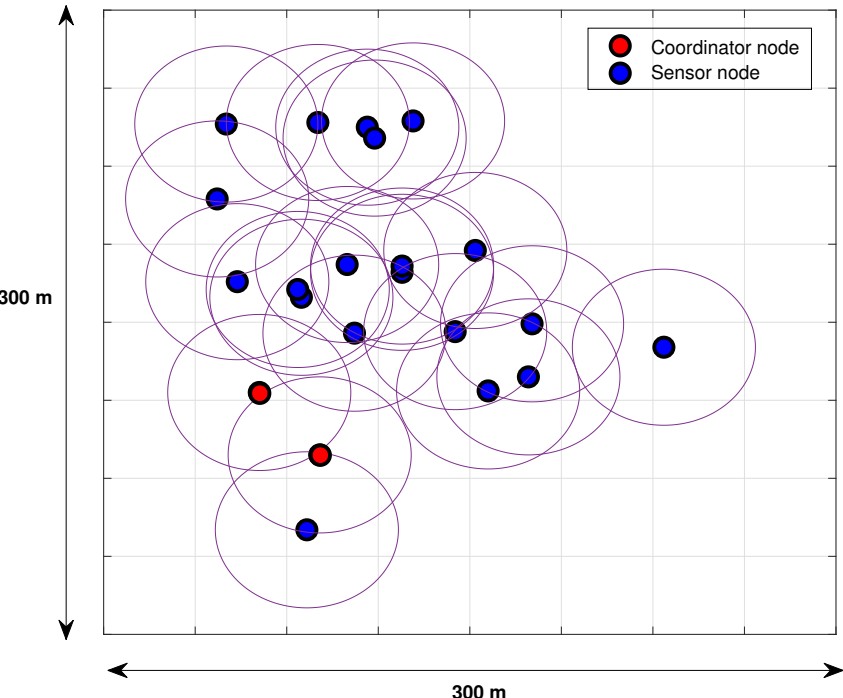

**Figure 17.** Distribution of nodes in the test space.

Figure 18 shows an example of the plot of the packets with some wireless technology, in this case, LoRa. The package and console display interface is www.thethingsnetwork.org, which allows us to observe the quality parameters of links, frequencies, lost packets, etc. Thanks to this analysis and the energy model, we can sharpen the precision of the model and further refine the types of energy to detect possible inefficient expenses of actions in the network nodes.

```
 1  {
 2    "gw_id": "eui-c0ee40ffff294455",
 3    "payload": "QNAfASaAHAABSf7SJiI=",
 4    "f_cnt": 28,
 5    "lora": {
 6      "spreading_factor": 7,
 7      "bandwidth": 125,
 8      "air_time": 46336000
 9    },
10    "coding_rate": "4/5",
11    "timestamp": "2019-08-13T22:19:27.036Z",
12    "rssi": -43,
13    "snr": 9.25,
14    "dev_addr": "26011FD0",
15    "frequency": 904700000
16  }
```

**Figure 18.** Example of frame for performance metrics.

Table 6 shows three useful performance metrics for analyzing the behavior of a network. The results for three different types of protocol are described: reagent (AODV), hybrid (MPH), and proactive (ZTR). The overall performance is better in the MPH protocol because it has route redundancy yet, it does not have so many routes to generate too much overhead and network collisions. We also

note that the simulator has the ability to accurately reproduce (approximately 2% difference) the real scenario under the specific conditions on each node according to the wireless technology used. In this way, different routing protocol rules suitable for coexisting networks in a wireless medium can be tested and energy optimization models can be generated according to the technology used and the target application. This makes the simulator an effective tool in predicting packet routing and power consumption models.

**Table 6.** Performance metrics for the real scenario and the simulation.

| Metric | Real Scenario | | | Simulation | | |
|---|---|---|---|---|---|---|
| | **AODV** | **MPH** | **ZTR** | **AODV** | **MPH** | **ZTR** |
| **Energy (J)** | 265.87 | 140.51 | 166.64 | 262.37 | 142.94 | 160.73 |
| **Delay from the furthest node (s)** | 1.9456 | 1.1567 | 1.0123 | 1.8955 | 1.1166 | 1.0573 |
| **PDR (%)** | 72 | 90 | 81 | 70 | 88 | 79 |

Using this simple energy model, we can implement energy saving techniques in some of the activities carried out by the nodes and we are capable of quantifying their impact on the total expenditure. For Figure 19, we implemented the MPH protocol in order to contrast the energy expenditure of each type of energy in the model under both stable and adverse conditions. For the four different wireless technologies, the rules of the MPH hybrid protocol were used. We observed that when there were adverse conditions such as shutting down, 15% of the nodes were turned off for 10 min every two hours, implying that the energies that have the greatest consumption impact are: CSMA energy, transmission energy, and receiving energy, increasing their value approximately by 40%. In these kind of situations, the model allows the use of routing protocols suitable for the operation of the network.

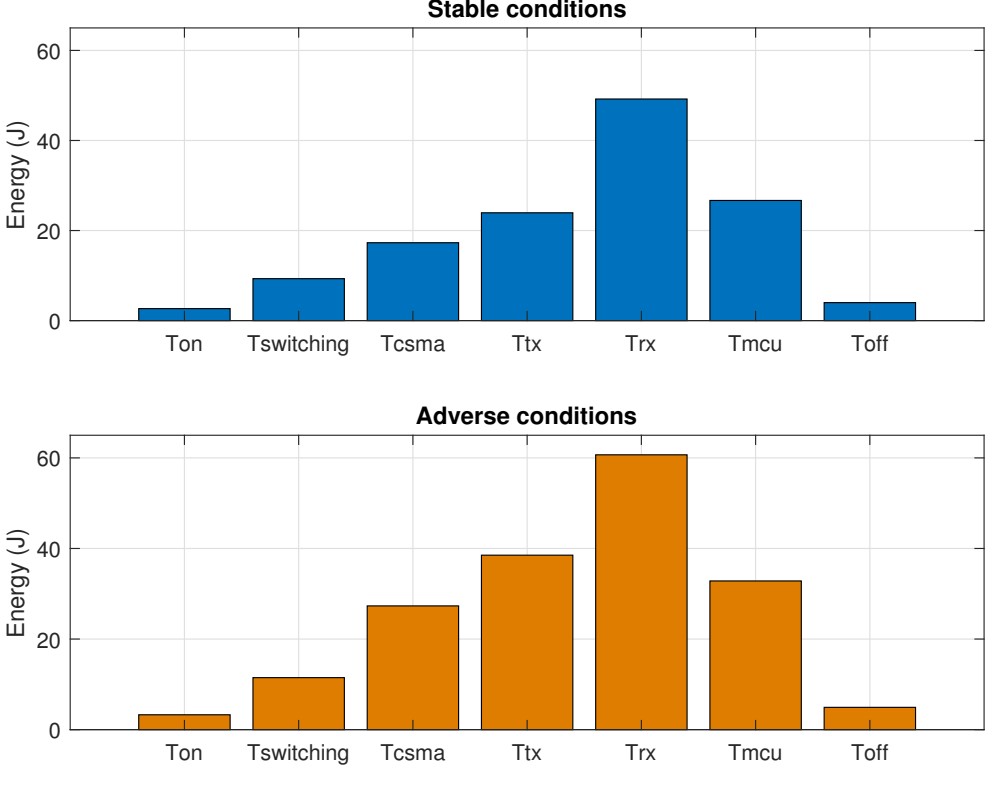

**Figure 19.** Energy under stable and adverse conditions for 22 nodes for each type of the energy model.

### 3.2. Extrapolation of the Experimental Scenario

The experiments carried out with 22 sensors were then expanded to 100 sensors and 200 sensors. This is in order to predict the results of the energy model by means of an event-based simulator, programmed in C ++ and previously tested in [10].

Next, it is interesting to analyze how the energy model behaves when the network begins to grow in its number of nodes. Table 7 shows three representative protocols out of the six we have studied in this work. Two predominant types of energies were analyzed based on the scalability of the network, from 22 to 300 nodes. Simulations were run for a full day and for nodes which parameters are detailed in Table 3.

**Table 7.** Energy model scalability for AODV, PEGASIS, and MPH protocols.

| Protocol | 22 nodes | 100 nodes | 150 nodes | 200 nodes | 250 nodes | 300 nodes |
|----------|----------|-----------|-----------|-----------|-----------|-----------|
| AODV | $E_{RX} = 151.35J$ | $E_{RX} = 301.73J$ | $E_{RX} = 403.23J$ | $E_{RX} = 567.85J$ | $E_{CSMA} = 425.02J$ | $E_{CSMA} = 431.86J$ |
| | $E_{TX} = 186.84J$ | $E_{TX} = 298.77J$ | $E_{TX} = 404.11J$ | $E_{TX} = 572.45J$ | $E_{TX} = 703.42J$ | $E_{TX} = 721.33J$ |
| PEGASIS | $E_{RX} = 99.32J$ | $E_{RX} = 125.23J$ | $E_{RX} = 200.61J$ | $E_{RX} = 301.53J$ | $E_{CSMA} = 342.55J$ | $E_{CSMA} = 415.55J$ |
| | $E_{TX} = 86.23J$ | $E_{TX} = 98.45J$ | $E_{TX} = 110.48J$ | $E_{TX} = 144.67J$ | $E_{TX} = 368.26J$ | $E_{RX} = 402.56J$ |
| MPH | $E_{RX} = 73.94J$ | $E_{RX} = 101.45J$ | $E_{RX} = 192.37J$ | $E_{RX} = 265.11J$ | $E_{RX} = 325.38J$ | $E_{RX} = 404.88J$ |
| | $E_{TX} = 90.46J$ | $E_{TX} = 111.52J$ | $E_{TX} = 193.44J$ | $E_{TX} = 203.78J$ | $E_{TX} = 382.31J$ | $E_{TX} = 440.03J$ |

Results in Table 7 show the two predominant energies for each network and for each of the three protocols studied. Here, we highlight the utility of the proposed model, which breaks down the total energy into the basic energy types of a node. This information becomes relevant when analyzing the skeleton of what is happening in the node with respect to performance metrics such as overhead, collisions, packet loss, interference, complexity of handling neighbor or routing tables, etc.

Analyzing the data, we observe that, when the network is large (about 300 nodes), the AODV and PEGASIS protocols present predominantly CSMA and TX energies. This may suggest that PEGASIS is not a protocol for networks that are too large because the route chain to reach the destination node becomes complex and packets can be lost. Below the 200 nodes, the three protocols under evaluation present the energies of RX and TX as preponderant. Visualizing in detail the network of 100 nodes, the difference between AODV and MPH is 67 % in favor of MPH, and the difference between MPH and PEGASIS is 58% in favor of MPH for the reception energy. This is an example that shows that MPH has a hierarchical topology and that the neighbor tables of the nodes become more manageable thanks to the periodicity of their update. The AODV protocol has the highest energy expenditure and this may be due to the fact that it has a greater number of control packets and if the routes expire or become obsolete, the nodes must start the entire route request process again. In this case, when the networks increase in size, the proposed energy model is very useful because, according to the characteristics of the network, it can establish weak points of behavior and detect possible gaps in energy loss. For example, if CSMA power is increased too much, this may be showing that the communication channel is continuously busy or that this is combined with multiple packet retransmissions, which implies that the processing time increases and the performance of the device is impaired. Then, the breakdown of energies performed by the model can detect abnormal increases or decreases in the behavior of the nodes, as occurs, for example in the PEGASIS protocol when the network is very large.

## 4. Conclusions

Unlike other types of wireless networks, Wireless Sensor Networks (WSNs) involve low-cost and low-processing devices, which send information to a collector node or base station (coordinator node). Due to the small size of nodes, the saving of energy consumption is vital since it is very difficult to recharge batteries and these networks aim to achieve maximum efficiency in the delivery of information in the harshest environments.

This paper has proposed the design and implementation of a simple and easy to develop energy model, which aim is to observe locally and globally the energy of the nodes in a network under almost any routing protocol. It has been verified that this model yields clear and concrete results for the main tasks performed by a node in the network. Based on this analysis, it could detect some anomalous behavior and know exactly in which phase of execution a problem is happening.

Another fundamental contribution is the analysis of performance metrics, not so common or obvious, because during the transmission of information, the processing of routes, and the observation of the means of communication, the nodes present valuable information when some strange behavior is occurring and this is immediately reflected in metrics such as: retransmissions of packets, listener retries to the communication channel, delays, overload of control packets, hop numbers, valid routes to a destination, among others. These metrics are directly related to energy and to the application of the proposed model. Changes or different behaviors are clearly observed in specific areas of the network.

The MPH protocol works very well in terms of processing, efficient information delivery, and low energy consumption, maintaining route redundancy. The detailed energy model shows that AODV and DSR have route backups and extensive routing tables, therefore a node can reach almost any point in the network. The model also shows that ZTR has low redundancy, is prone to failures, and has a small number of valid routes. However, it is simple, fast, and consumes a small amount of energy. The combination of a hierarchical topology with auto-configuration mechanisms and maintenance of the MPH protocol makes the nodes capable of optimizing network processes, reduce delays by up to 25%, take short routes to the destination, and reduce network overload even in a 30%. All this is reflected in the successful delivery of information. In addition, the proposed model allows us to understand that between MPH and LEACH and PEGASIS there is only a difference of 3% and 2% energy savings for the last two protocols. Thus, the model analyzes the energy impact of each type of energy for optimization of the algorithm in various protocols of the literature [5–10]

**Author Contributions:** C.D.-V.-S. developed the energy algorithm, built the simulator, prepared and executed the simulations, interpreted and analyzed the results, designed the methodology and drafted the manuscript. C.M.-P. supervised the research methodology and the approach of this work, he performed the formal analysis. J.A.N.-F. reviewed, interpreted and drafted the simulation results, he also strongly contributed to the design of the energy scheme. R.V. was involved in the analysis of the energy model under routing protocols and he run validation, he reviewed the methodology and the manuscript. A.R.-S. worked in the formal analysis and the manuscript. All authors have read and agreed to the published version of the manuscript.

**Funding:** This research received no external funding.

**Conflicts of Interest:** The authors declare no conflict of interest.

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
