# Peer review of "Wireless Sensor Network Energy Model and Its Use in the Optimization of Routing Protocols"

_energies, doi:10.3390/en13030728_

Round 1

Reviewer 1 Report

This paper has proposed an energy model to observe the energy of the nodes under almost any routing protocol. The article is well written and it has enough contribution. Especially, the authors have clarified important parts of the proposed model. Based on this, I will recommend the publication of this revised version.

Author Response

Thank you very much for Reviewer’s comments and his/her observations were of vital importance for the development of this research and great contributions to our knowledge.

Reviewer 2 Report

The reviewer appreciates the efforts that the author put in this manuscript. The quality of this paper is greatly improved in comparison with previous version.

The reviewer suggest the authors to compare the performance of the proposed MPH routing protocol with recently proposed energy-aware routing protocols in the literature. Comparing the proposed routing MPH protocol with non-energy-aware routing protocols such as AODV, DSR, ZTR is not appropriate.

It is recommended that the authors should improve the organizing structure of this manuscript. The authors should present Sections 1~4 concisely and remove redundant parts/sections if necessary. It is suggested that the summary of energy-aware routing protocols is given first, followed by the motivations of proposing the MPH routing protocol. Then, details of the operations of the proposed MPH routing protocol should be presented. The simulation results should focus on the performance comparison of the proposed MPH routing protocol with recently proposed energy-aware routing protocols in the literature. For each evaluated performance, the comparison results should be plotted on the same graphs.

The font size of Algorithm 1 should be increased to enhance its readability. 

Author Response

Dear Editor

Energies Editorial Office

We are submitting the revised version of our paper:

“Wireless Sensor Networks Energy Model and its Use in the Optimization of Routing Protocols”

Authored by: Carolina Del-Valle-Soto, Carlos Mex-Perera, Juan Arturo Nolazco-Flores, Ramiro Velázquez, Alberto Rossa-Sierra

We would like to thank the reviewers and editors for their detailed analysis of the manuscript; the comments were very valuable to us. In the revised version, we have incorporated the all changes recommended by the reviewers.

Comments to all observations and suggestions including point-by-point responses are addressed in the following text.

_______________________________________________________________________________

Reviewer 2 comments

Comment 1: The reviewer appreciates the efforts that the author put in this manuscript. The quality of this paper is greatly improved in comparison with previous version.

The reviewer suggest the authors to compare the performance of the proposed MPH routing protocol with recently proposed energy-aware routing protocols in the literature. Comparing the proposed routing MPH protocol with non-energy-aware routing protocols such as AODV, DSR, ZTR is not appropriate.

Response: Thank you very much for Reviewer’s comments and his/her observations were of vital importance for the development of this research and great contributions to our knowledge.

We would like to comment to the Reviewer that his observation is very pertinent. However, we would like to emphasize that the MPH protocol is a hybrid protocol, that is, it is a combination of both reactive and proactive nature protocols. The AODV and DSR protocols are two protocols widely recognized in WSN for their rationality and the ZTR protocol is a proactive protocol by excellence. Therefore, we believe that the comparison between protocols of same nature is relevant. In addition, we wanted to complement our study with energy-aware protocols because the main objective of this work is to demonstrate and analyze the higher energy costs in the sensors, according to the type of tasks they perform on the network.

Two of our previous publications address a feasible comparison between the protocol we study in the paper (AODV, DSR, ZTR, and MPH):

 [1] Del-Valle-Soto, C., Mex-Perera, C., Orozco-Lugo, A., Lara, M., Galván-Tejada, G., & Olmedo, O. (2014). On the MAC/Network/Energy performance evaluation of wireless sensor networks: contrasting MPH, AODV, DSR and ZTR routing protocols. Sensors, 14(12), 22811-22847.

[2] Del-Valle-Soto, C., Mex-Perera, C., Monroy, R., & Nolazco-Flores, J. (2017). MPH-M, AODV-M and DSR-M performance evaluation under jamming attacks. Sensors, 17(7), 1573.

In addition, the presentation, explanation, analysis and comparison of the MPH protocol was previously reported in our previous work:

Del-Valle-Soto, C., Mex-Perera, C., Orozco-Lugo, A., Galván-Tejada, G. M., Olmedo, O., & Lara, M. (2014, April). An efficient multi-parent hierarchical routing protocol for WSNs. In 2014 Wireless Telecommunications Symposium (pp. 1-8). IEEE.

Nevertheless, we thank the Reviewer for his comment. In consequence, we have improved the depth of the state of the art in protocols that could complement each other according to their nature and we have added the following analysis (Lines 65-91).

There are also other protocols analyzed in the literature, such as Cluster based Energy Efficient Location Routing Protocol (CELRP), which is a hierarchical protocol with nodes distributed in clusters and arranged in quadrants. Each quadrant contains two clustering, which would be like the master nodes, and other nodes transmit data with two hops data transmission. Another similar protocol is Position Responsive Routing Protocol (PRRP), which exhibits a better energy efficiency. This protocol makes a choice of the cluster head based on distance from the sink, energy level, and the average distance of neighboring nodes from the candidate master node. PRRP is similar to the LEACH protocol in which any node communicates with the sump and the data transmission mechanism is the time-based schedule. In PRRP, the number of nodes of the branches of the hierarchical tree and the distance from the non-leaf node is smaller compared to LEACH and CELRP. This makes energy conservation optimization possible [11]. The PRRP protocol dramatically increases data transfer and provides a better solution to the routing problem focused on energy efficiency, due to the efficient selection and distribution of gateways. Another important protocol to mention is the Energy-Efficient data Routing Protocol for WSN (EERP) [12], which selects a set of good roads, and chooses one based on the node state and the road cost function. In EERP, each node has a number of neighbors through which packets can be routed to the base station. A node bases its routing decision on two metrics: status and cost function.

We added two more references:

[11] Zaman, N.; Tang Jung, L.; Yasin, M.M. Enhancing energy efficiency of wireless sensor network through the design of energy efficient routing protocol. Journal of Sensors 2016, 2016.

[12] Ghaffari, A. An energy efficient routing protocol for wireless sensor networks using A-star algorithm. Journal of applied research and technology 2014, 12, 815–822.

Comment 2: It is recommended that the authors should improve the organizing structure of this manuscript. The authors should present Sections 1~4 concisely and remove redundant parts/sections if necessary. It is suggested that the summary of energy-aware routing protocols is given first, followed by the motivations of proposing the MPH routing protocol. Then, details of the operations of the proposed MPH routing protocol should be presented. The simulation results should focus on the performance comparison of the proposed MPH routing protocol with recently proposed energy-aware routing protocols in the literature. For each evaluated performance, the comparison results should be plotted on the same graphs.

Response: Thank you very much for Reviewer’s comments.

The Reviewer is right and we have added a paragraph in the Introduction, in this way we have eliminated the redundant sections, as follow:

This paper proposes a simple energy model, which quickly evidences changes in the network performance, its implementation is simple, and it does not represent higher processing consumption.  In the Wireless Sensor Networks literature, there are few energy models [11], [12] and some energy-aware protocols that seek to optimize the energy of networks. The need for an energy model that influences the performance metrics of a network is an advantage that not all models exhibit. This indicates that we can know, according to each type of task that the node performs, the major and minor impact on parameters such as: resilience, overhead, packet retransmissions, listening retries to the communication channel, delay, and many others.

It is important to keep in mind that the objective of this work is not to present the MPH protocol, because this protocol was already presented and supported in previous works as shown in the previous comment. We have clarified the motivation because the Reviewer is right that the objective was not very clear. In this way, we want to compare the energy analysis in several routing protocols known in WSN in order to demonstrate the analysis in each of the types of energy according to the basic tasks of a sensor in the network. Here we present comparative graphs from Figures 4 to 11 and later, we describe a real scenario with different wireless technologies to analyze a case study on a university campus.

Besides, we added a paragraph in the motivation:

This energy model is characterized by being simple and evidencing the energy used by the main tasks of a node in the network. In addition, it reflects the impact of each type of energy on the performance metrics of a network, with which, we could optimize routing protocols according to the conditions of the network and its local expenditure (at node level).

We have summarized in the Introduction the relevant paragraphs of the Related Work section, as recommended by the Reviewer. In addition, the following sections link perfectly with the state of the art that is now mentioned in the Introduction. Moreover, we have combined the "Scope of Routing protocols in sensor networks" and "Significance for Study of Energy in Wireless Sensor Networks" sections and converted them into subsections of the Introduction. In addition, we have added the paragraphs of Global Energy and Local Energy (Lines 364-373): as part of the subsection dealing with energy.

Comment 3: The font size of Algorithm 1 should be increased to enhance its readability. 

Response: Thank you very much for Reviewer. He/she is right and we have increased the font of the algorithm.

Thank you very much.

Sincerely,

Carolina Del Valle Soto

Universidad Panamericana. Facultad de Ingeniería. Álvaro del Portillo 49, Zapopan, Jalisco, 45010, México.

Phone: +52 (33) 13682200 | Ext. 4245

Reviewer 3 Report

Here authors have proposed a WSN energy model for calculating energy at each node for various routing protocols. There are a number of points that authors need to address:

Authors should review the use of qualifiers. For example: Introduction: Lines 54 and 55: "...this work becomes notorious": what do authors mean by this?

There are a number of run off sentences throughout the manuscript: sentences that are too long (longer than 2 lines). Please split those into smaller sentences.

Introduction: There is very little mention of previous work. Please explain in brief (expand on the lines 60 and 61).

Please explain in introduction the need to propose another energy model. 

The motivation section just summarizes the work presented in the manuscript. It does not talk about the motivation or why this work was needed? What gap does this work fill in the current compendium of energy consumption related models.

Authors should provide the code for the model either in supplementary materials or a publicly accessible repository.

Section 2 is very long and since this manuscript is not a review paper, a brief summary of section 2 can be integrated appropriately with the Introduction. 

There are no proper transition sentences between the sections.

In fact, authors should integrate section 3 with the Introduction as well. General field introduction, background, previous work, motivation, scope and significance should all be in a 2-3 page introduction . Please summarize and integrate appropriately.

Authors jump from past and present tenses throughout the manuscript. Please proof read for sentence formation and other grammatical errors.

Line 369-374 seem to be a repetition of the previous paragraph. Please edit the manuscript to remove repetitions. 

Please provide a table of assumptions made for the model and the positive and negative controls for energy consumption conditions.

Fig3 and algorithm 1 can be integrated into a single figure.

Have authors looked into optimization of CSMA/CA algorithm execution frequency for each node? Please discuss in the manuscript.

What happens when the packet transmission rate is varied? please discuss.

What about CELRP protocol and Position Responsive Routing Protocol (PRRP)? Please evaluate and compare in the manuscript.

What happens if the bluetooth nodes operate on 5GHz? Please discuss.

For lines 845 and 846 please include references/citations.

Author Response

Dear

Editor

Energies Editorial Office

We are submitting the paper:

“Wireless Sensor Networks Energy Model and its Use in the Optimization of Routing Protocols”

Authored by: Carolina Del-Valle-Soto, Carlos Mex-Perera , Juan Arturo Nolazco-Flores , Ramiro Velázquez , Alberto Rossa-Sierra

We would like to thank the reviewers and editors for their detailed analysis of the manuscript; the comments are very valuable to us. In the revised version of thepaper,we have incorporated the all changes recommended by the reviewers.

Comments to all observations and suggestions including point-by-point responses are addressed in the following text.

_______________________________________________________________________________

Reviewer 3 comments

Comment 1: Here authors have proposed a WSN energy model for calculating energy at each node for various routing protocols. There are a number of points that authors need to address:

Authors should review the use of qualifiers. For example: Introduction: Lines 54 and 55: "...this work becomes notorious": what do authors mean by this?

Response: Thank you very much for your comments and we agree with them.These comments have contributed to the improvement of the paper. Now, we have changed some sentences, for example: “This is where the contribution of this work becomes relevant: the proposed model quickly evidences changes in the network performance, its implementation is simple, and it does not represent higher processing consumption”. In order to clarify why the proposed energy model represents an impact in the analysis of performance metrics in various routing protocols, with easy implementation.

Comment 2: There are a number of run off sentences throughout the manuscript: sentences that are too long (longer than 2 lines). Please split those into smaller sentences.

Response: Thank you very much for your comments and we have corrected the paragraphs too long and paid special attention to the writing of the sentences throughout the paper.

Comment 3: Introduction: There is very little mention of previous work. Please explain in brief (expand on the lines 60 and 61).

Please explain in introduction the need to propose another energy model. 

Response: Thank you very much for your comment.

We have increased the state of the art and why we have chosen these protocols to make comparisons with the proposed energy model.

This paper intends to test the proposed energy model and observe its repercussions on proactive and reactive sensor network protocols. The analysis is described quantitatively by observing the performance metrics that will positively or negatively affect such model. This is where the contribution  of this work becomes relevant: the proposed model quickly evidences changes in the network  performance, its implementation is simple, and it does not represent higher processing consumption.  The proposed model is then compared against the performance of network sensors under some widely known protocols: Ad hoc On demand Distance Vector (AODV) [5], Dynamic Source Routing (DSR) [6], ZigBee Tree Routing (ZTR) [7], Low Energy Adaptive Clustering Hierarchy (LEACH) [8] and, Power Efficient Gathering in Sensor Information Systems (PEGASIS) [9]. These protocols will also be compared against the Multi-Parent Hierarchical (MPH) routing protocol proposed, designed, and implemented by the authors in a previous work [10]. In this study, AODV, DSR, ZTR, LEACH and, PEGASIS are quantitatively compared and assessed based on several efficiency metrics that analyze how these routing protocols optimize energy through various schemes in order to find the best routes in the shortest possible time. As the hierarchy algorithms, such as the ZTR, denote simple and fast routing that reduce network overloads, they are reliable and have a distributed addressing scheme that only permits neighbor tables, not long, and elaborated routing tables. The performance of wireless sensor networks is closely related to that of the routing protocol, because routes can vary dynamically over time. Energy-aware protocols such as LEACH and PEGASIS seek to increase the life time of the network. They propose to find sub-optimal paths to allow a more equitable distribution of the network’s energy consumption. Hierarchical protocols such as ZTR and MPH have advantages in terms of scalability and efficiency in communications. Particularly for WSN networks, nodes with higher energy can be used to process and send information, while those with lower energy are used to monitor the environment and send the information to the node with greater energy capacity. Finally, proactive type protocols, which establish routes before there is a real traffic demand, are suitable for real-time traffic, since they have low latency, however, they waste bandwidth due to periodic updates and they are not energy efficient.

In addition, we have added two references for two more energy-aware protocols: LEACH and PEGASIS.

Bhat, G.; Sreenivasan, A. REVIEW ON ENERGY OPTIMIZATION AND CLUSTER BASED ROUTING PROTOCOL IN WSN 2019. Trigunait, C.K.; Prabha, S. A Novel Energy Efficient Security Protocol In WSN. International Journal of Information Technology (IJIT) 2019.

The need to propose another energy model is explained here:

This paper proposes a simple energy model, which quickly evidences changes in the network performance, its implementation is simple, and it does not represent higher processing consumption.  In the Wireless Sensor Networks literature, there are few energy models [11], [12] and  some energy-aware protocols that seek to optimize the energy of networks. The need for an energy  model that impacts the performance metrics of a network is an advantage that not all models present. This indicates that we can know, according to each type of task that the node performs, what is the major and minor impact on parameters such as: resilience, overhead, packet retransmissions, listening retries to the communication channel, delay, and many others.

And we added two more references:

Ordónez, F.; Krishnamachari, B. Optimal information extraction in energy-limited wireless sensor networks. 897 IEEE Journal on Selected Areas in Communications 2004, 22, 1121–1129.

Zhou, H.Y.; Luo, D.Y.; Gao, Y.; Zuo, D.C. Modeling of node energy consumption for wireless sensor 899 networks. Wireless Sensor Network 2011.

Comment 4: The motivation section just summarizes the work presented in the manuscript. It does not talk about the motivation or why this work was needed? What gap does this work fill in the current compendium of energy consumption related models.

Response: Thank you very much. The Reviewer is right and we have added a paragraph in the Introduction, as follow:

This paper proposes a simple energy model, which quickly evidences changes in the network performance, its implementation is simple, and it does not represent higher processing consumption.  In the Wireless Sensor Networks literature, there are few energy models [11], [12] and  some energy-aware protocols that seek to optimize the energy of networks. The need for an energy  model that impacts the performance metrics of a network is an advantage that not all models present. This indicates that we can know, according to each type of task that the node performs, what is the major and minor impact on parameters such as: resilience, overhead, packet retransmissions, listening retries to the communication channel, delay, and many others.

Besides, we added a paragraph in the motivation:

This energy model is characterized by being simple and evidencing the energy used by the main tasks of a node in the network. In addition, it reflects the impact of each type of energy on the performance metrics of a network, with which, we could optimize routing protocols according to the conditions of the network and its local expenditure (at node level).

Comment 5: Authors should provide the code for the model either in supplementary materials or a publicly accessible repository.

Response: Thank you very much. The Reviewer can see an example code of the equations from (1) to (8) for each node under Zigbee, LoRa and BLE technologies. We added supplementary materials, and we show you here some principal codes for running the energy model:

First, we present the application in https://console.thethingsnetwork.org/applications

For BLE:

For LoRa:

For Zigbee:

The simulation and core codes of the sensors are owned by Monster Engineering, Electronics Engineer & Tgo®.

Comment 6: Section 2 is very long and since this manuscript is not a review paper, a brief summary of section 2 can be integrated appropriately with the Introduction. 

Response: Thank you very much. We have summarized in the Introduction the relevant paragraphs of the Related Work section, as recommended by the Reviewer. In addition, the following sections link perfectly with the state of the art that is now mentioned in the Introduction.

Comment 7: There are no proper transition sentences between the sections.

Response: Thank you very much for your comments and we have corrected some paragraphs and paid special attention to the writing of the sentences throughout the paper.

Comment 8: In fact, authors should integrate section 3 with the Introduction as well. General field introduction, background, previous work, motivation, scope and significance should all be in a 2-3 page introduction. Please summarize and integrate appropriately.

Response: Thank you very much for your comment. We have combined the "Scope of Routing protocols in sensor networks" and "Significance for Study of Energy in Wireless Sensor Networks" sections and converted them into subsections of the Introduction. In addition, we have added the paragraphs of Global Energy and Local Energy as part of the subsection dealing with energy.

Comment 9: Authors jump from past and present tenses throughout the manuscript. Please proof read for sentence formation and other grammatical errors.

Response: Thank you very much for your comments and we have corrected some paragraphs throughout the paper.

Comment 10: Line 369-374 seem to be a repetition of the previous paragraph. Please edit the manuscript to remove repetitions. 

Response: We agree with the Reviewer and we rewrite the two paragraphs:

In this paper, an analytical model is proposed and exemplified by the operation parameters of the Texas Instruments CC2530 chip [33], which has a radio interface as per the IEEE 802.15.4 standard. This model shows the energy expended for each of the activities performed from the moment a node is added to a network and becomes part of it, listens to the channel, receives and sends messages, executes the link layer algorithms data, and changes states, ending with the energy consumed when it turns off and disconnects from the network. The energy used by the microcontroller depends on the operation mode. For example, techniques for turning nodes off reduce energy consumption by setting the microcontroller in idle mode for certain time intervals [37]. However, for this analysis, it is assumed that the mentioned chip in each node operates in continuous active mode at 32 MHz at the microcontroller’s clock frequency to better study how energy consumption behaves under a routing protocol determined without the influence of techniques used for turning nodes off. Thus, the total energy used by the microcontroller will be given by (1).

Comment 11: Please provide a table of assumptions made for the model and the positive and negative controls for energy consumption conditions.

Response: Thank you very much. We have set Table 1 to show the average operating values of the CC2530 sensor, as model parameters for each chip, according to the manufacturer. The energy performance of the chip is shown in the reference. This document describes software examples for the System-on-Chip solution. It also describes the necessary hardware and software to run the examples, and how to get started.

Besides, for nodes under LoRa technology and WiFi, we use the model described in [34]

Bouguera, T.; Diouris, J.F.; Chaillout, J.J.; Jaouadi, R.; Andrieux, G. Energy consumption model for sensor 934 nodes based on LoRa and LoRaWAN. Sensors 2018, 18, 2104.

Comment 12: Fig3 and algorithm 1 can be integrated into a single figure.

Response: Thank you very much. The Reviewer is absolutely right and we have left algorithm 1 in the manuscript, because it presents in more detail the energy model. We have left the explanation of the flowchart to clearly describe equations 1 to 8 in more detail for any given node "i".

Comment 13: Have authors looked into optimization of CSMA/CA algorithm execution frequency for each node? Please discuss in the manuscript.

Response: Thank you very much. Many thanks to the Reviewer for his comment and we have analyzed in detail the CSMA/CA algorithm. In the simulator we are using, we use the MAC-level protocol that is used from all extensions of 802.15.4 (including the original version), which is the CSMA / CA, which guarantees a high data rate. A network recognition is being carried out at all times to check the status of the channel (carrier detection). Only when free, can the data be sent. In the 802.11 standard, the physical layer polls the energy level over the radio frequency to determine whether or not there is transmission. If the channel is busy, start a random timer (with a maximum of five backoff periods), the timer only discovers time with free channel, transmits when it expires, and finally, if it does not receive ACK increases the backoff.

We have discussed the above in the manuscript based on the frequency of execution of the algorithm.

Comment 14: What happens when the packet transmission rate is varied? please discuss.

Response: Thank you very much. We added more tests, as follow:

Regarding the rate of transmitted packets, we analyze what the measurement impact of the energy model when we vary the packet transmission rate. Tests are carried out for 1 day (24 hours), specifically they were carried out on Wednesday in order that on the campus there was a regular traffic of people and vehicles, therefore, more wireless devices.

We note that, for five different packet transmission rates, the most drastic variation is found in the rates of 50 and 100 kbps, with a 37% difference with respect to energy at 250 kpbs. The difference in energy expenditure with the change in packet transmission rates is 43% higher for reactive protocols, compared to other protocols. And the difference between proactive protocols and energy-aware protocols is only 5% in favor of energy-aware protocols.

Comment 15: What about CELRP protocol and Position Responsive Routing Protocol (PRRP)? Please evaluate and compare in the manuscript.

Response: Thank you very much to the Reviewer. Here our discussion:

There are also other protocols analyzed in the literature, such as Cluster based Energy Efficient Location Routing Protocol (CELRP), which is a hierarchical protocol with nodes distributed in clusters and arranged in quadrants. Each quadrant contains two clustering, which would be like the master nodes, and other nodes transmit data with two hops data transmission. Another similar protocol is Position Responsive Routing Protocol (PRRP) WSN routing protocol which is more energy efficient. This protocol makes a choice of the cluster  head based on distance from the sink, energy level, and the average distance of neighboring nodes from the candidate master node. PRRP is similar to the LEACH protocol in which any node can communicate with the sump and the data transmission mechanism is the time-based schedule. In PRRP, the number of nodes of the branches of the hierarchical tree and the distance from the non-leaf node is smaller compared to LEACH and CELRP. This makes energy conservation can be optimized [11]. The PRRP protocol dramatically increases data transfer and provides a better solution to the routing problem focused on energy efficiency, due to the efficient selection and distribution of gateways. Another important protocol to mention is the Energy-Efficient data Routing Protocol for wireless sensor networks (EERP) [12], which selects a set of good roads, and chooses one based on the node state and the road cost function. In EERP, each node has a number of neighbors through which packets can be routed to the base station. A node bases its routing decision on two metrics: status and cost function.

Besides, we added two more references:

Zaman, N.; Tang Jung, L.; Yasin, M.M. Enhancing energy efficiency of wireless sensor network through the design of energy efficient routing protocol. Journal of Sensors 2016.

Ghaffari, A. An energy efficient routing protocol for wireless sensor networks using A-star algorithm. Journal of applied research and technology 2014, 12, 815–822.

Comment 16: What happens if the bluetooth nodes operate on 5GHz? Please discuss.

Response: Thank you very much to the Reviewer. Here our discussion:

Bluetooth is a wireless communication link, operating in the unlicensed ISM band at 2.4 GHz using a frequency hopping transceiver. This frequency band is 2400 - 2483.5 MHz. According to the standard (https://www.mouser.it/pdfdocs/bluetooth-Core-v50.pdf), our BLE modules are focused on streaming audio, not video, which operates in the unlicensed ISM (Industrial Scientific Medical) band at 2.4 GHz. However, there are modules that are quad band, which have Bluetooth and WiFi and therefore say they can reach 5 GHz (referring to WiFi). So, the Bluetooth SIG maintains regulatory content associated with Bluetooth technology in the 2.4 GHz ISM band.

The 5 GHz band has a smaller range compared to the 2.4 GHz band, in radio frequencies, the higher frequency, the smaller range. If we use a lower frequency such as 2.4 GHz, the distance it will cover will be greater than that of the 5 GHz band.

It may also be that the Reviewer has suggested consulting us about the Bluetooth 5 version. This is the fifth version of this popular wireless communication protocol that, traditionally, has served both to transfer files between two devices, and to transmit audio. Currently, versions of Bluetooth devices do not have the ability to be backwards compatible with previous versions.

Comment 17: For lines 845 and 846 please include references/citations.

Response: Thank you very much to the Reviewer. Here our modification of those lines:

Thus, the model analyzes the energy impact of each type of 886 energy for optimization of the algorithm in various protocols of the literature [5], [6], [7], [8], [9], [10].

Thank you very much.

Sincerely,

Carolina Del Valle Soto

Universidad Panamericana. Facultad de Ingeniería. Álvaro del Portillo 49, Zapopan, Jalisco, 45010, México.

Phone: +52 (33) 13682200 | Ext. 4245

Round 2

Reviewer 2 Report

The reviewer thanks the authors for your effort into answering the reviewer's comment. Since, the authors have addressed all previous comments, the reviewer would like to recommend this manuscript to be published in present form. 

Reviewer 3 Report

Authors have appropriately addressed the comments. As a future comment,  making the Introduction and background sections more succinct can make the manuscript more effective in conveying the impact.

This manuscript is a resubmission of an earlier submission. The following is a list of the peer review reports and author responses from that submission.

Round 1

Reviewer 1 Report

In this paper, the authors introduce an energy model which tries to cover the global and local energy consumption activities in wireless sensor networks. The reviewer appreciate the efforts that the author put in this manuscript. However, there are many flaws in this manuscript that need to be addressed.

- Major issues:

1. The energy model shown in Fig. 2 may not cover all the energy consumption activities in wireless sensor networks. More importantly, before using the proposed energy model for further investigations, the authors should present some results to verify the accuracy of the proposed energy model in predicting the total energy used by each node. In addition, putting t_switching in P(mW) axis seems does not make sense.

2. The benefits of using the proposed energy model is ambiguous. The authors did not clearly show the advantage of this proposed model in the considered MPH and other well-known routing protocols. Instead, the presentation in Section 4 rather show the calculation of energy consumption of node in the network.

3. Subsection 4.2 is not suitable to be put in Section 4. Moreover, there is a lack of summarizing previous energy aware routing protocols in the literature.

4. How was the value of the energy per node obtain in Fig. 5? Is it obtain from a one simulation run or from taking average of many simulation runs? From the viewpoint of the reviewer, the later method is more appropriate.

5. As mentioned above, since the accuracy of the proposed energy model was not verified, the reliability of all later results is in doubt. Furthermore, there was no comparison with previously proposed energy aware routing protocols in the literature. Comparing with the energy consumption of AODV, DSR, ZTR is unfair.

6. Instead of presenting the percentage of valid routes of each routing protocol, the authors should use the packet delivery ration (PDR) or throughput, which is a common metric for evaluating the quality of routing protocol.

7. In Subsection 5.1, the authors should give the map of testing area and the location of each node on this map. Giving Fig. 9 and Fig. 11 separately is not appropriate. Moreover, a photo of the implemented network system (with each component) should be provide instead of presenting like Fig. 10.

Minor issues:

1. "where" in many paragraphs should be aligned appropriately.

2. Citation of Table 1 is missing.

3. "eq. 1" should be changed to "(1)". The same rule is applied for other equation citations.

5. The font size of Algorithm 1 in small, thus, it content can not be seen clearly. Moreover, "Algoritmo 1" should be corrected as "Algorithm 1".

5. Line 438: "300 m2" should be corrected as "300 m x 300 m"; Line 439: "using the topology shown" is confusing phrase.

6. The legend for Fig. 8 should be given.

Reviewer 2 Report

The article proposes a novel model to optimise the energy consumption of nodes in a WSN. The model is well defined by employing appropriate figures and examples. Furthermore, the performance of the proposed model is evaluated in a real-world scenario. I will recommend the publication of this article after addressing the following issues:

In the abstract, in the first 5 lines, the novelty of the work is not explicitly mentioned. Authors have explained some features as a contribution which is too general and can be applied to any other similar proposed model in the literature. I suggest the authors rewrite this part by mentioning more specific contributions like what they did in lines 38 - 45.

In Algorithm 1, lines are too small. Please make them larger and also change the name "Algoritmo" to "Algorithm".

In Figure 8, it needs to be clear that which line represents the stable condition and which one represents the adverse condition.

Reviewer 3 Report

I appreciate the effort of the authors for putting this work together. However, the work has plenty flaws and can be improved. Here are some of my observation and queries

(1) Firstly, I hate to say, but this work is not well presented. The English and sentences do not flow. In fact it is very poorly written. I advise the work should be given to an English expert. Just may be this has affected my judgement of the paper (because I honestly struggle to understand the message the paper is carrying)

(2) The focus of this research is on the energy model and not on the existing routing protocol. Thus, the authors should focus on the energy model and compare their proposed energy model  with the most popular energy model used  in the literature. I mean the energy model used in 

An Application-Specific Protocol Architecture for Wireless Microsensor Networks.  (check equation 9 and 10).

Testing their energy model with existing routing protocol is less important and its deviate from the scope of the paper.

However, the authors can test  these existing routing protocols with their proposed energy model and the popularly used energy model to give remarkable conclusions and findings to strengthen the contribution of this paper.

(3) In Figure 1 and 4, some of the symbol/diagram used are not clearly defined or explained.

4) Based on the parameters given in Table 2, it is not clear how the total energy used by a node will be estimated from this proposed energy model. That is, given the packet length, what is the value assumed for I and V or how do we calculate the I and V for switching, receiving, Transmitting etc. Is it defined by Figure 2, If yes, the authors should explain clearly. 

5) Lastly, in the simulation test, the authors assumed 22 sensor nodes. This is very small. The authors should increase the number of nodes to 100 and above.